# Induced endosymbiosis between a fungus and bacterium reveals a shift from antagonism to commensalism

Thomas Gassler [1,3] ✉, Gabriel H. Giger [1], Anna Sintsova[1], Olivia X. Bossert [1], Alannah Holderbusch[1], Miriam Bortfeld-Miller[1], Benoit Dehapiot [2], Shinichi Sunagawa [1] & Julia A. Vorholt [1] ✉

Endosymbioses represent dynamic relationships between organisms that may involve antagonistic phases during their emergence. Here, we induced cell-in-cell interactions between the free-living bacterium *Ralstonia pickettii* and an endosymbiont-free strain of the fungus *Rhizopus microsporus* using fluidic force microscopy to investigate the early phase of endosymbiosis formation. Following the implantation of bacteria into the cytosol, the rapid proliferation of *R. pickettii* compromised host fitness, as evidenced by reduced fungal viability, and triggered immune responses characterized by upregulated expression of stress-related defense genes. Vertical transmission of bacteria across fungal generations enabled repeated rounds of selective passaging, ultimately resulting in transcriptional relaxation of the fungal defense response. High-throughput-imaging revealed that the propagated system accommodated higher bacterial loads within viable spores, with a corresponding reduction in fungal growth. The observed physiological changes and comparative fungal transcriptomic profiles indicated adaptive resilience and a shift from antagonism to commensalism. This transition was characterized by attenuated expression of genes involved in cell wall remodeling and reactive oxygen metabolism. Our experimental system provides insights into the early processes of endosymbiosis, supporting the hypothesis that facultative intracellular pathogens can serve as intermediates toward stable endosymbiotic relationships.

Endosymbioses, where one organism lives within another, exist along a spectrum of interactions ranging from parasitic to mutualistic. Although many established endosymbioses are mutualistic, this need not be the initial driving force at their origin[1,2]. Antagonistic interactions are increasingly recognized as potential starting points for long-term endosymbioses[3-8]. Intracellular pathogens possess inherent capabilities that predispose them to endosymbiotic transitions. These include mechanisms for host cell invasion, intracellular survival and immune evasion[9-12]. Such adaptations may provide selective advantages during the early stages of endosymbiosis, potentially followed by attenuated virulence and the expansion from horizontal to vertical transmission.

Computational and mathematical models corroborate the evolutionary trajectory from pathogenicity to commensalism or mutualism, indicating that stable interactions can emerge from initially conflictual ones when vertical transmission and host tolerance co-evolve[12-14].

[1]Institute of Microbiology, ETH Zurich, Zurich, Switzerland. [2]ScopeM, ETH Zurich, Zurich, Switzerland. [3]Present address: Department of Bioengineering and Bezos Centre for Sustainable Protein, Imperial College London, London, United Kingdom. ✉e-mail: t.gassler@imperial.ac.uk; jvorholt@ethz.ch

Despite these theoretical frameworks, empirical demonstrations capturing this transition, particularly at its onset, remain limited, in part due to the challenge of accessing and maintaining cell-in-cell associations between organisms that are typically free-living[2].

Early-diverging fungal hosts represent a model system to study the emergence of endosymbiosis, as they comprise both naturally endosymbiont-harboring and endosymbiont-free strains[15,16]. In particular, *Rhizopus microsporus* has emerged as a powerful model system for studying endosymbiosis[17,18], and its interplay between bacterial natural product formation[19–23] and fungal defense mechanisms[23]. Within this fungal species, intracellular bacteria are found only in specific host-symbiont combinations, with the majority being endosymbiont free[15]. Natural endosymbiont harboring (EH) host strains of *R. microsporus* exhibit an obligate dependency on their bacterial endosymbiont *Mycetohabitans rhizoxinica* for sporangiospore formation, while endosymbiont-free (non-host, NH) strains can reproduce by asexual sporulation[24]. The interaction is considered mutualistic: the fungal host benefits from bacterial produced secondary metabolites that enhance fitness by promoting nutrient acquisition and protection from predation[23,25], while the endosymbiont gains access to intracellular nutrients and vertical transmission through fungal spores[26]. This established symbiosis relies on specialized bacterial mechanisms, including the Type 2 secretion system (T2SS) that facilitates invasion via secretion of cell-wall degrading enzymes[27] and a Type 3 secretion system (T3SS) that delivers effector proteins critical for establishing and maintaining the symbiosis[5,28]. Despite these invasion mechanisms, numerous *R. microsporus* strains effectively resist bacterial invasion and maintain an endosymbiont-free lifestyle[5,29].

To investigate the emergence of novel endosymbiotic relationships in *R. microsporus*, we previously developed a targeted delivery strategy that bypasses natural entry barriers, thereby allowing for defined starting points of intracellular encounters[30] while preserving host viability. To overcome the rigid cell wall, we adapted the fluidic force microscopy (FluidFM) technology to inject bacteria directly into the cytosol of both EH and NH strains of *R. microsporus*[30]. The injection of different bacterial strains yielded markedly different outcomes. Implantation of *M. rhizoxinica* into the NH strain resulted in initially rare events of vertical transmission to spores and an improved fitness of the system through adaptive laboratory evolution. In contrast, transfer of *Escherichia coli* led to rapid proliferation of the bacterium and triggered a fungal immune response, forming non-viable septa-enclosed compartments with bacteria, and viable bacteria-free compartments. These outcomes highlight the critical importance of compatibility between the microbial lifestyles[30].

In this study, we chose the ecologically versatile[31] *Ralstonia pickettii* K-288 as a bacterial candidate for transplantation into the axenically sporulating[24] NH strain. This strain selection was made to establish a new intracellular pairing between a non-host fungal species and a free-living bacterial strain, lacking a natural endosymbiotic relationship. *R. pickettii* shares phylogenetic proximity with *Mycetohabitans*[32], while maintaining a predominantly free-living lifestyle across diverse environmental niches including water and soil habitats[33]. Though occasionally identified as an opportunistic pathogen associated with contaminated medical devices[34], *R. pickettii* also demonstrates potential for beneficial associations with hosts. As an intracellular symbiont, *R. pickettii* reduces phagocytosis of spores by macrophages and contributes to host virulence, as described for a clinical isolate of *R. microsporus*[35]. These observations suggest that *R. pickettii* can persist within a fungal cytoplasm, making it a promising candidate to probe the emergence of a synthetic endosymbiosis.

We hypothesized that *R. pickettii*, when introduced into the cytosol of a NH strain, may initially be perceived as a pathogen but might, under positive selection, transition toward a more stable association. Following the injection of fluorescently labeled *R. pickettii*

into germlings of the NH strain, we tracked the pairings across cellular and generational levels, using transcriptome profiling to characterize molecular responses. Bacterial impact on host fitness changed over the course of time. Deep-learning assisted imaging analysis revealed a decrease in growth rates of bacteria-colonized fungi upon serial passaging, while surviving spores tolerated a higher bacterial load. The observed fungal phenotypic shifts were reflected at the transcriptional level: the early interaction phase exhibited strong upregulation of fungal cell wall remodeling and reactive oxygen species (ROS)-associated genes, while later interaction phases showed a partial reversion to wild-type expression profiles, including attenuation of immune-like responses, suggesting adaptive resilience of the fungus under selection for maintaining the interaction.

## Results

### Injection of *R. pickettii* into *R. microsporus* and vertical transmission

We first tested whether the *R. pickettii* K-288 type strain, labeled with mCherry, can infect the EH or NH strain of *R. microsporus* under laboratory conditions (Supplementary Fig. 1). Neither the cured EH strain after antibiotic-induced removal of *M. rhizoxinica* nor the mycelium from the NH strain showed observable colonization by *R. pickettii* under the examined conditions. We further checked whether *R. pickettii* can co-infect the cured EH strain when the natural endosymbiont *M. rhizoxinica*, labeled with GFP, enters through the cell wall. Following co-incubation experiments of the cured fungus, the mycelium started to sporulate and the asexual sporangiospores (referred to as spores) either contained *M. rhizoxinica* ($57 \pm 15\%$ Gfp positive spores) or lacked bacteria, and no mCherry positive spores were detectable. Similarly, for the NH strain, only bacteria-free spores were recovered after exposure to the bacterial mix, indicating that *R. pickettii* is not able to naturally infect the intracellular compartments of *R. microsporus*.

Following our recently established workflow[30], we next performed FluidFM based injection of mCherry labeled *R. pickettii* into the NH strain, to establish a de novo cell-in-cell interaction between a free-living bacterium and non-host fungus. Upon injection of *R. pickettii* into the cytosol of the fungus both partners showed consistent growth (Fig. 1a–d). The introduced bacteria resumed growth in the cytosol after an initial lag phase of up to 6 h in three independent experiments (see Supplementary Video 1 for cytosolic growth of a single injected bacterium). The intracellular doubling time of the bacteria was about 1.7 h and thus similar to in vitro growth rates in liquid medium and on agar (Fig. 1e and f, and Supplementary Fig. 2). As fungal growth progressed, we noticed the formation of bacteria-rich zones inside the mycelium (Supplementary Fig. 3a–e). In contrast to injection with *E. coli*, where protective septa formation was commonly observed[30], *R. pickettii* rich areas were only occasionally enclosed by septa. After recovery on a plate for 5 days, pronounced growth by mCherry expressing *R. pickettii* was visible in the mycelium (Supplementary Fig. 3c, d). The bacteria were seemingly displaced passively along cytosolic mass flow within growing hyphae, which could explain why bacteria-rich zones often formed at apical regions where cytoplasmic streaming is most pronounced due to active hyphal extension (Supplementary Video 2 and Supplementary Fig. 3f). After recovery and growth on agar plates, the injected fungus maintained its ability to sporulate.

We then used fluorescence activated cell sorting (FACS) to analyze and sort the asexual spores from three individual injections of *R. pickettii* into *R. microsporus* NH and found that a small fraction (i.e., 0.009%, 0.9% and 1.2%) of the total population was positive for the bacteria-associated mCherry signal (see population $B_{pos}$ in Fig. 1g as example). The presence of *R. pickettii* inside the spores was verified by confocal microscopy (Fig. 1h). After transfer to liquid growth medium, a subset of spores germinated and the bacteria recommenced

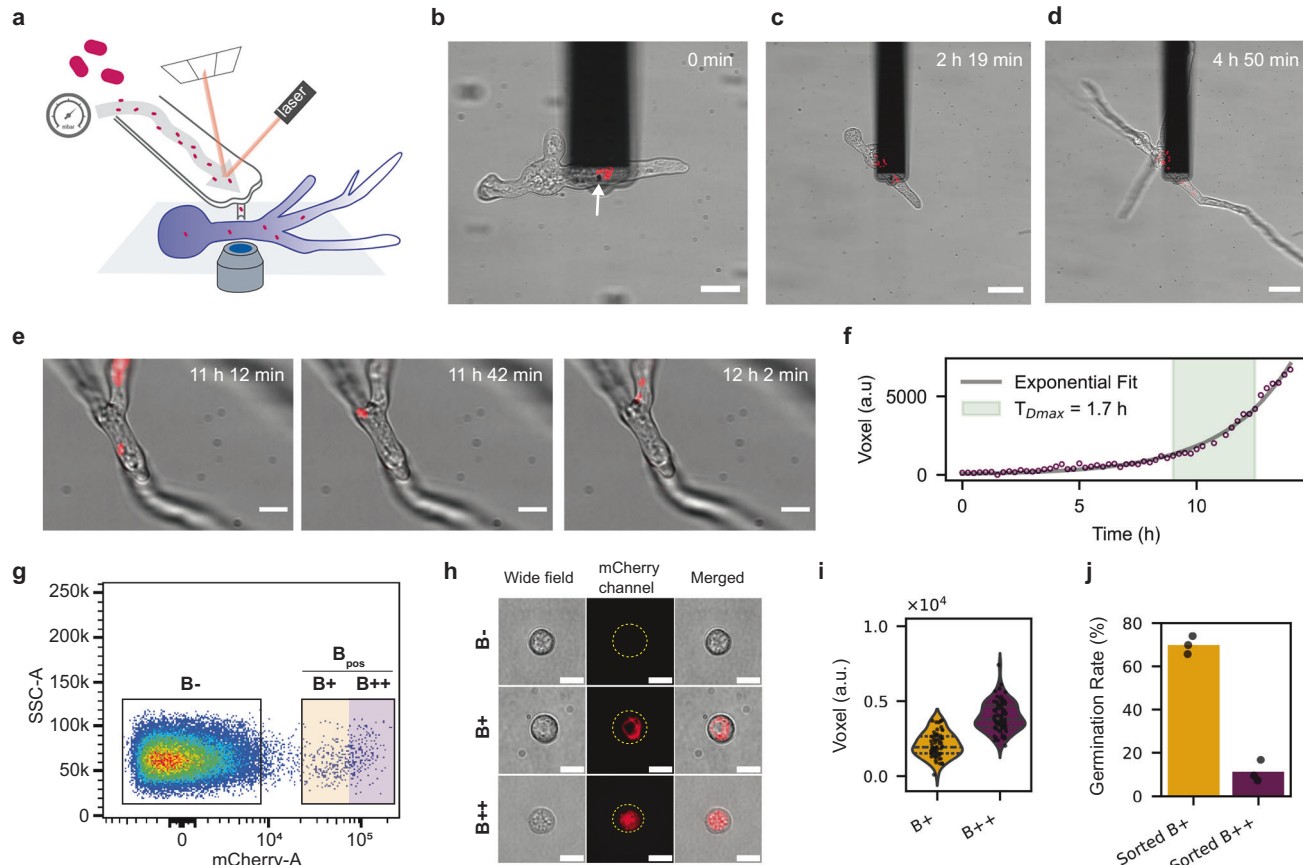

**Fig. 1 | Injection of *Ralstonia pickettii* into *Rhizopus microsporus* and vertical transmission. a** Scheme of FluidFM injection of mCherry-expressing *R. pickettii* cells into *R. microsporus* NH-strain germlings on a glass dish surface placed above an inverted confocal light microscope. The internal turgor of the hypha is overcome by applying overpressure up to 6.5 bar. **b-e** Images of germling and *R. pickettii* cells at time points after the FluidFM injection; merged widefield and mCherry channel. **b** Image shows injected bacterial cells in red next to the puncture site (white arrow), the experiment was repeated independently four times with similar results. **c** Injected germling starts to recover after isolation into a fresh dish and bacteria start to migrate along mass flow of the cytosol. **d** Injected germling and bacteria start to grow. **e** Time series on a single *R. pickettii* cell shows division of the cell. **f** The intracellular maximum doubling time ($T_{Dmax}$) of *R. pickettii* after injection is approximately 1.7 h; for determination a single bacterial cell was followed by timelapse microscopy directly after injection and imaged for 15 h, the volume of bacteria was expressed as Voxel (a.u.) using a Matlab based script (see Supplementary Fig. 2 and ref. 60), gray line shows an exponential fit over all datapoints and time where maximum growth is observed is highlighted in green. **g** Flow cytometry plot for spores harvested from an injected fungus which was transferred onto agar plate; shown are the negative spore population (B-) and mCherry positive spore populations (B+ and B++). The fractions of bacteria positive cells after injection were 0.009, 0.9 and 1.2% in three independent experiments. **h** Images of FACS-sorted B-, B+ and B++ spores with intracellular bacteria after vertical transmission. The experiment was carried out three times with similar results. **i** Violin plot showing the bacterial volume in Voxel (a.u.) per spore for positive sorted fractions B+ and B++. At least 15 spores were analyzed by confocal microscopy for each condition. **j** Germination success is dependent on bacterial load. Spores were collected at round four after injection from a B++ line and sorted into B++ and B+. Germination success was determined by counting growing germlings in 96-well plates after two days of inoculation; the unit of study was a single spore (one spore per well); for each group (parental B++, sorted B++, sorted B+), $n = 288$ spores (3 × 96-well plates). Shown are the mean (bars) and individual plate values from three technical replicates per group (one 96-well plate = one technical replicate, filled with 96 spores from the respective population). For b,c,d and h images show a single-z-layer wide-field image overlaid with the two-dimensional projection of the mCherry-signal z-stack in red. e shows an overlay of two-dimensional projections of the wide-field and mCherry-signal z-stack. Scale bars, 10 μm (**b**), 40 μm (**c**, **d**), 10 μm (**e**) and 5 μm (**h**). Source data are provided as a Source Data file and imaging data on zenodo (ref. 69).

intracellular growth (Supplementary Video 3), confirming their successful vertical transmission in all three injections.

To correlate the bacterial load with germination success, we divided the bacteria-positive spore population ($B_{pos}$) into low (B+, those in the lower 50% of the positive population) and high (B++, those in the upper 50% of the positive population) bacterial load categories (exemplified in Fig. 1g and Supplementary Fig. 4). Fluorescence microscopy of the FACS-sorted fractions confirmed an increased bacterial load in B++ over B+ spores (Fig. 1h and i). The germination success of these spores was inversely correlated with the bacterial load (Fig. 1j). This reflected a load-dependent fitness cost on the fungal host, consistent with the impairment previously documented in the NH strain after injection of *M. rhizoxinica*[30].

## Experimental selection affects host fitness

To stabilize the cell-in-cell system, we designed a serial passaging experiment that commenced with injection and cultivation to sporulation, sorting of spores according to bacterial load, and subsequent re-cultivation. We anticipated that sorting according to bacterial load (B+ and B++, Fig. 1g) would act as an adaptive filter, probing the adaptability and robustness of the NH strain as a new host system. For every cultivation round, between 50,000 and 100,000 spores were sorted and transferred to solid medium to induce germination. In each subsequent passaging round, the sorting criterion according to the previous one (i.e., B+ or B++) was maintained. By applying this defined selection pressure, we followed both sorting lines (B+ and B++) from each of the three independent injections for five rounds, extending

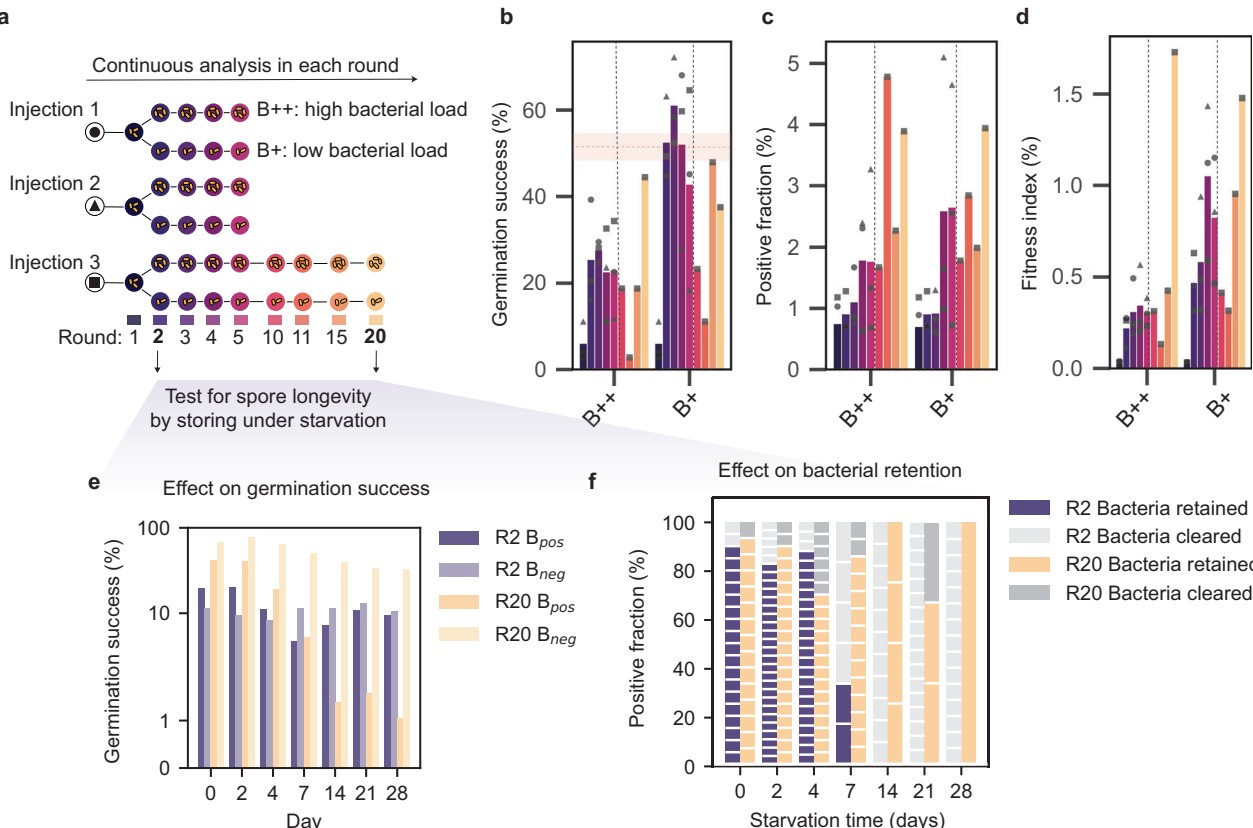

**Fig. 2 | Continuous passaging using positive selection affects host fitness and resilience towards starvation. a** Scheme for the serial passaging experiment: three individual injections of *R. pickettii* into the non-host (NH) train were performed, followed by five rounds of FACS based selection according to high (B++) and low (B+) bacterial load. Injection three was adapted for 20 rounds along B++ and B+ regime. **b–d** Fitness increases over time in adaptation. In each round (colored to match scheme in (**a**), germination success (**b**), fraction of positive spores (**c**) and fitness index (**d**) were determined as measure for holobiont fitness. Fungal spores from three individual injections were propagated for five rounds and one injection for 20 rounds according to high (B++) and low (B+) selection regime each; bars before vertical dashed line show average of three biological replicates (circle = injection 1, triangle = injection 2, square = injection 3), bars after vertical line show value from injection 3. The red horizontal dashed lines and shading indicates the germination success of negative spores from the non-injected NH strain. **e, f** The passaged lineage exhibits increased stability against nutrient depletion, maintaining bacterial load during germination (color code of rounds corresponding to (**a**). **e** Germination success of positive (B+) and negative (B-) spores after storage in buffer for up to 28 days. Germination success was obtained

by counting growing germlings in 96-well plates. The spores were stored for the indicated days (*x* axis) at 16 °C and activated by adding rich medium and cultivating for two days at 28 °C before germination was quantified. The bars show the germination success after different days of storage (R2 B+ , day 0: 37 from 192 spores germinated, day 2: 38/192, day 4: 32/288, day 7: 9/192, day 14: 14/192, day 21: 21/192, day 28: 18/192; R2 B-, day 0: 22/192, day 2: 18/192, day 4: 8/96, day 7: 22/192, day 14: 22/192, day 21: 4/288, day 28: 20/192; R20 B+ , day 0: 121/288, day 2: 117/288, day 4: 55/288, day 7: 15/288, day 14: 4/288, day 21: 3/288, day 28: 1/288; R20 B-, day 0: 192/ 288, day 2: 222/288, day 4: 182/288, day 7: 146/288, day 14: 114/288, day 21: 97/288, day 28: 93/288). **f** Growing germlings were evaluated microscopically for presence of bacteria. Depicted are counts of the tested germlings originating from bacteria positive spores (R2 B+ and R20 B+ , white stacked lines of each bar indicate the number of germlings in each sample that were verified by microscopy. Dark colored bars show the count of bacteria positive germlings (R2 Bacteria Positive and R20 Bacteria Positive) and light-colored bars the counts of bacteria negative ones (R2 Bacteria Cleared and R20 Bacteria Cleared). Source data are provided as a Source Data file and FACS data on zenodo (ref. [69]).

both lines from one injection to 20 rounds (schematically represented in Fig. 2a).

The germination success of bacteria-positive spores (B$_{pos}$) in round one was 3.8%, 11.1%, and 2.8% for the three individual injections, respectively. This was considerably lower than the baseline of 52 ± 6.5% for the non-injected NH strain (Fig. 2b, and Supplementary Fig. 5a). Between the first and second rounds, both B++ and B+ spores showed a sharp increase in germination success, which then plateaued from rounds 2 to 5, albeit with fluctuations specific to each injection. In parallel, we recorded the proportion of bacteria-positive spores in each round and observed an increase in spore colonization by *R. pickettii* across all six replicates (three independent injections, each passaged according to B+ and B++ regime). This trend was reflected in an increasing fitness index (calculated as the product of germination success and the fraction of spores that harbor *R. pickettii*), indicating a progressive improvement of the merged system (Fig. 2c, d).

To assess the dynamics of the interaction, we extended one of the injection lines for 15 additional passaging rounds, both for the B++ and B+ regimes (Fig. 2a). We observed a drop in germination success at round 10 and 11, which recovered by round 15. Concurrently, the germination success of bacteria-negative spores followed a similar trend, indicating a systemic fitness deficit (Supplementary Fig. 5c). When normalizing the germination success of bacteria-positive spores to those of negative spores in each round, we observed that the fitness of the merged system increased initially (from round one to round two) and then plateaued during serial passaging (Supplementary Fig. 6). Following a decline in rounds 10 and 11, both B++ and B+ conditions exhibited an increasing germination success and a rise in the proportion of bacteria-positive spores. The B+ selection line, which maintained a lower bacterial load, showed superior fitness to B++ throughout the entire passaging rounds. In addition to examining temporal dynamics, we quantified the colonization of *R. pickettii* in the

mature mycelium of the B+ line and compared it to *M. rhizoxinica* in its native EH and evolved NH background[30]. *M. rhizoxinica* colonization ranged from one bacterial copy per approximately 900 to 2400 fungal actin copies (EH, one *ACT1* copy in genome) and one per 2700 to 16000 (NH, two *ACT1* copies in genome), while *R. pickettii* reached one per two to 15 fungal copies (see Supplementary Fig. 7). These results reveal that *R. pickettii* proliferates up to two orders of magnitude more than the natural endosymbiont intracellularly.

## Starvation resilience and bacterial retention changes during serial passaging

To investigate the effect of the bacterial colonization on the system, we subjected spores from both early (round 2) and late (round 20) stages of the serial passaging experiment to nutrient starvation conditions (Fig. 2e, f). We used spores from cryo stocks and cultured them once on a plate to restore their physiological state. Freshly generated single spores were sorted into individual wells containing buffer without a carbon source and stored at 16 °C to assess their germination success and the presence of bacteria upon germination.

Under starvation conditions, bacteria-negative ($B_{neg}$) spores germinated more successfully than bacteria-positive ($B_{pos}$) spores for R20 (Fig. 2e). Both R2 and R20 $B_{pos}$ spores reduced their initial germination success by approximately 50% (57% and 45% respectively) after 4 days of nutrient starvation. However, the R20 $B_{pos}$ spores had twice the germination success of the R2 $B_{pos}$ spores at the beginning of the experiment, demonstrating the benefits of prolonged passaging.

We verified the presence of *R. pickettii* in the germlings by fluorescence microscopy (Fig. 2f). Under starvation conditions, the two lines, R2 and R20, showed contrasting patterns of bacterial retention. In R2 germlings, the proportion harboring bacteria declined progressively from initial levels to 33% by day 7, with bacterial signals becoming undetectable at all subsequent timepoints (days 14, 21, and 28). Conversely, R20 germlings consistently maintained bacterial signals in over 65% of individuals across all sampled timepoints throughout the 28-day starvation period. These findings suggest that the R20 line acquired an enhanced tolerance to nutrient limitation while maintaining its bacterial load, whereas the ancestral line appeared to clear *R. pickettii* during prolonged nutrient starvation.

## Phenotypic trade-offs and host tolerance to bacterial load

The dynamic fitness patterns and the nutrient starvation response (Fig. 2) assessed during continued passaging of the host-bacteria pairing revealed a phenotypic transition in the fungal host. This shift progressed from an initial bacterial intolerance to progressively higher tolerance levels, indicating a change in the relationship from pathogenic to increasingly commensal.

To further study the interaction at the single-cell level, we developed a deep learning-based procedure to segment and track individual spores over time. This method enabled high-throughput monitoring of fungal growth rates and host responses using multi-positional microscopy (Fig. 3 and Supplementary Fig. 8).

Next, we analyzed spores from round 2, 11 and 20 of the serial passaging experiment. Out of the 384 individually tracked spores, 114 started to germinate. Their intracellular bacteria then resumed growth and occasionally appeared to be sequestered in vacuole-like compartments (Supplementary Video 4). To estimate fungal growth rate, we first extracted the fungal area as proxy for fungal volume of the germinating spores over a 16 h time course (see Supplementary Fig. 8). Fungal biomass doubled at comparable rates for the NH-strain (98 ± 13.5 min) and the round 2 derived NH-strain samples (94 ± 5.5 min). After 11 and 20 rounds of passaging, however, doubling times lengthened to 113 ± 38 min (ns) and 112 ± 23 min ($p = 0.014$, Kruskal–Wallis test followed by Dunn's post-hoc test with Bonferroni correction) revealing a deceleration in biomass accumulation (Fig. 3b). The germination success values during imaging were lower for the

R2 samples than for more adapted spores from round 11 and 20 (Fig. 3c). We further examined the effect of the bacterial load of the spores on the germination success for the different samples (Fig. 3d). Spores with a high bacterial load in round 2 samples were not able to germinate, while round 11 and round 20 spores accepted a higher bacterial burden, as the overall bacterial load of growing spores increased 2.3 fold in round 11 and 2.1 fold in round 20 compared to the round 2 samples ($p$ values 0.00005 and 0.00613 respectively, Mann–Whitney U test vs. R2). These results indicated that *R. pickettii* colonized round 11 and round 20 spores more densely, while the fungus became more resilient towards the associated bacterial burden.

## Temporal dynamics of fungal transcriptional responses during prolonged cell-in-cell interaction

Building on our bacteria injection and passaging experiment, we conducted a longitudinal analysis of transcriptional responses of the NH strain fungus at two distinct rounds (rounds 10 and 20). This analysis focused on the B+ line, which demonstrated a consistently high fitness index in rounds 15 and 20 and could be reliably revived from frozen stocks. First, we compared the non-injected NH-strain with bacteria-positive germlings after round 10 (R10 $B_{pos}$, positive population after 10 rounds of adaptation under low bacterial load (B+) selection) and round 20 (R20 $B_{pos}$, positive population after 20 rounds of adaptation under low bacterial load (B+) selection) of the serial passaging experiment. Principal component analysis (PCA) revealed distinct global gene expression profiles among these groups (Fig. 4a and Supplementary Fig. 9). Notably, R20 $B_{pos}$ samples clustered more closely with the NH-strain than R10 $B_{pos}$, indicating a progression toward wildtype-like transcriptional behavior and adaptive resilience.

Differential gene expression analysis revealed more transcriptional alterations compared to NH strain at round 10 (1210 upregulated, 567 downregulated genes) than at round 20 (463 upregulated, 405 downregulated; adjusted $p$ value < 0.05, fold change >1.5; Fig. 4b). The common subset of differentially expressed genes (DEGs) maintained consistent directionality of change across both rounds (Fig. 4c), despite representing different functional categories according to GO-term analysis (Supplementary Data 1).

To identify key biological processes underlying bacterial accommodation by the host, we analyzed shared DEGs from both R10 $B_{pos}$ and R20 $B_{pos}$ samples compared to the NH strain (Fig. 4d and Supplementary Fig. 10). The analysis revealed consistent upregulation of genes related to cell wall synthesis and remodeling (Supplementary Data 2, Supplementary Fig. 10). Genes associated with iron homeostasis showed differential regulation at both passaging rounds, while genes involved in specific cell wall modifications, such as chitin deacetylases, mannosyltransferases, and septins, were prominently induced at R10 (ref. 36). Interestingly, R20 samples showed significant changes in RNA processing and methylation pathways not found in R10 (Supplementary Data 2). We also observed an enrichment of retrotransposable elements among differentially expressed genes at R20 (3.4% of all DEGs compared to 1.9% genome-wide) (20 out of 591 compared to 401 out of 21522), suggesting that these mobile genetic elements may facilitate host accommodation of intracellular bacteria through genome plasticity.

## Attenuation of fungal pathogenic responses during serial passaging

To dissect the temporal dynamics of gene expression in response to the presence of intracellular bacteria in different rounds of the same injection line, we conducted a direct comparison between R10 $B_{pos}$ and R20 $B_{pos}$ samples (Fig. 5). Genes upregulated at round 10, particularly those associated with cell wall remodeling (Supplementary Data 2), showed significant downregulation by round 20, indicating an attenuation of a defense response.

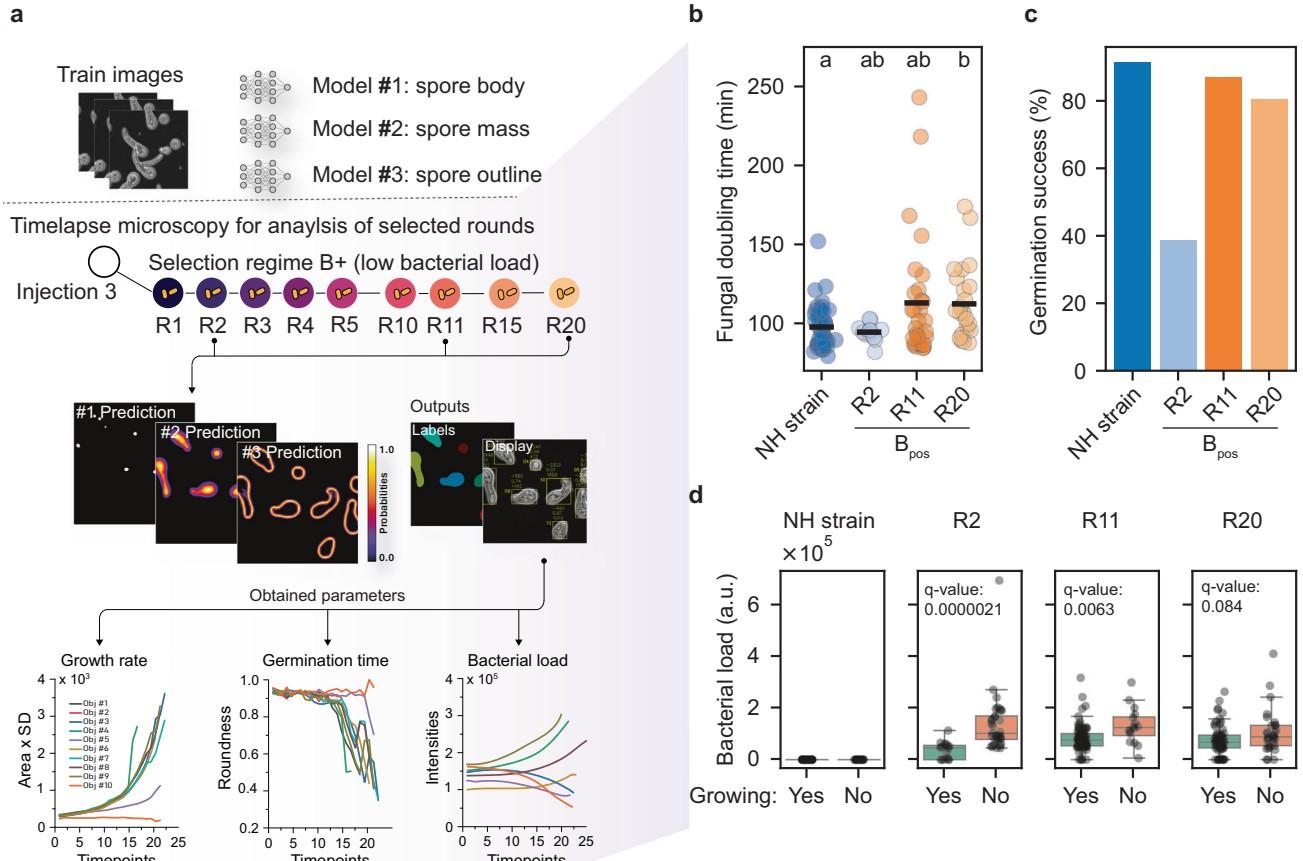

**Fig. 3 | Phenotypic trade-offs and host tolerance to bacterial load. a** Selected samples from injection three were characterized using a deep learning-based segmentation and tracking procedure. Different models to detect fungal body (#1), mass (#2) and outline (#3) detection were trained from annotated images (Supplementary Fig. 8); Spores from selected samples were regenerated and imaged using multi-point timelapse microscopy and analyzed in a semi-automated manner, the output parameters include area from a z-projection as a proxy for volume, a metric for germination state (Roundness) and the sum of fluorescence intensity within the detected fungal biomass as measure for bacterial load (Intensity). **b–d** Adapted spores exhibit a higher bacterial load, accompanied by reduced germination speed. Samples included rounds 2, 11, and 20 from the low bacterial load selection line (B+). Spores were derived from frozen B+ stocks by sorting mCherry-positive spores, cultivating on plates, and harvesting fresh spores. To ensure representation of the entire population, the harvested spores were sorted again for the entire mCherry positives ($B_{pos}$) before initiating imaging experiments. **b** The growth speed of positive spores decreases after serial passaging. NH strain and R2 B+ spores germinated faster than R11 and R20 B+ spores. Statistical analysis (Kruskal-Wallis test: Statistic = 10.59, $p$ value = 0.014, number of tracked / analyzed spores were 151/131 for the NH strain, 49/16 for R2 B+ , 92/76 for R11 B+ and 92/62 for R20 B+), followed by Dunn's post-hoc test with Bonferroni correction (two-

sided pairwise comparisons): NH strain vs R20 B+ adjusted $p$ = 0.016474; all other pairwise comparisons adjusted $p ≥$ 0.147053; groups that do not share a letter differ significantly at α = 0.05. **c** Germination success of adapted spores is higher than un-adapted spores. Germination success was obtained by dividing the number of germinating spores by the number of all detected spores in each condition: NH strain (138/151), R2 B+ (19/49), R11 B+ (74/92), R20 B+ (80/92). **d** Adapted spores accept a higher bacterial load. The bacterial load was estimated in all germinating (Yes) and non-germinating (No) spores (i.e., spores that doubled with a volume increase of 1.5 during the first 10 h were considered as germinating) as sum of the fluorescence intensity from all z-stacks (Bacterial Load (a.u.)) at beginning of the imaging experiment (two sided Mann-Whitney U test, $p$ values were false discovery rate (FDR) adjusted for multiple comparisons according to Benjamini–Hochberg ($q$ values): R2 B+ (16 germinating (Yes) and 33 non-germinating (No) spores, $p$ value < 0.0001, q = 2.127e-05), R11 B+ (76 Yes / 16 No, $p$ value = 0.0032, q = 0.00633), NH strain (131 Yes / 20 No, $p$ value = 0.816, q = 0.8155) and R20 B+ (62 Yes / 30 No, $p$ value = 0.063, q = 0.08356)); box plots depict the median (center line), the 25th to 75th percentiles (box), and whiskers indicate values within 1.5 times the interquartile range (IQR) of the quartiles; all individual observations are shown as jittered points and were included in the analyses. Source data are provided as a Source Data file.

The initial cell wall remodeling, characterized by chitin degradation, chitosan synthesis, and 1,3-beta-glucan branching, aligns with antagonistic reactions previously observed in another non-host *Rhizopus* strain when challenged with extracellular bacteria of the genus *Mycetohabitans*[5]. Our dataset identified 32 of the 44 previously described cell wall related defense genes (Supplementary Data 2). Except for three genes (annotated as septins and chitin deacetylases), the expression of these genes followed a temporal pattern that indicated a reduction in defense activity (Fig. 5b, green circles): robust induction during early stage of passaging (round 10) followed by reversion to wildtype-like expression in the later stage (round 20).

This temporal shift extended to known defense mechanisms involving reactive oxygen species (ROS). Six ROS-associated genes

previously identified (ref. 5) in bacterial challenge responses were differentially expressed between round 10 and 20 (Supplementary Data 2). In round 20 samples, we observed significant upregulation of a catalase and a heme peroxidase, an expression profile resembling that of natural host strains engaged in mutualistic interactions with *Mycetohabitans* (Supplementary Data 2 and ref. 5). Genome mining of the *R. pickettii* strain with AntiSMASH identified a biosynthetic gene cluster encoding rhizoferrin synthetase, a non-ribosomal peptide synthetase-independent siderophore (NIS), which potentially affects iron homeostasis and ROS production[37–39]. Remarkably, differential expression of these genes emerged only through temporal comparison of bacteria-positive samples, underscoring their nuanced regulation during serial passaging.

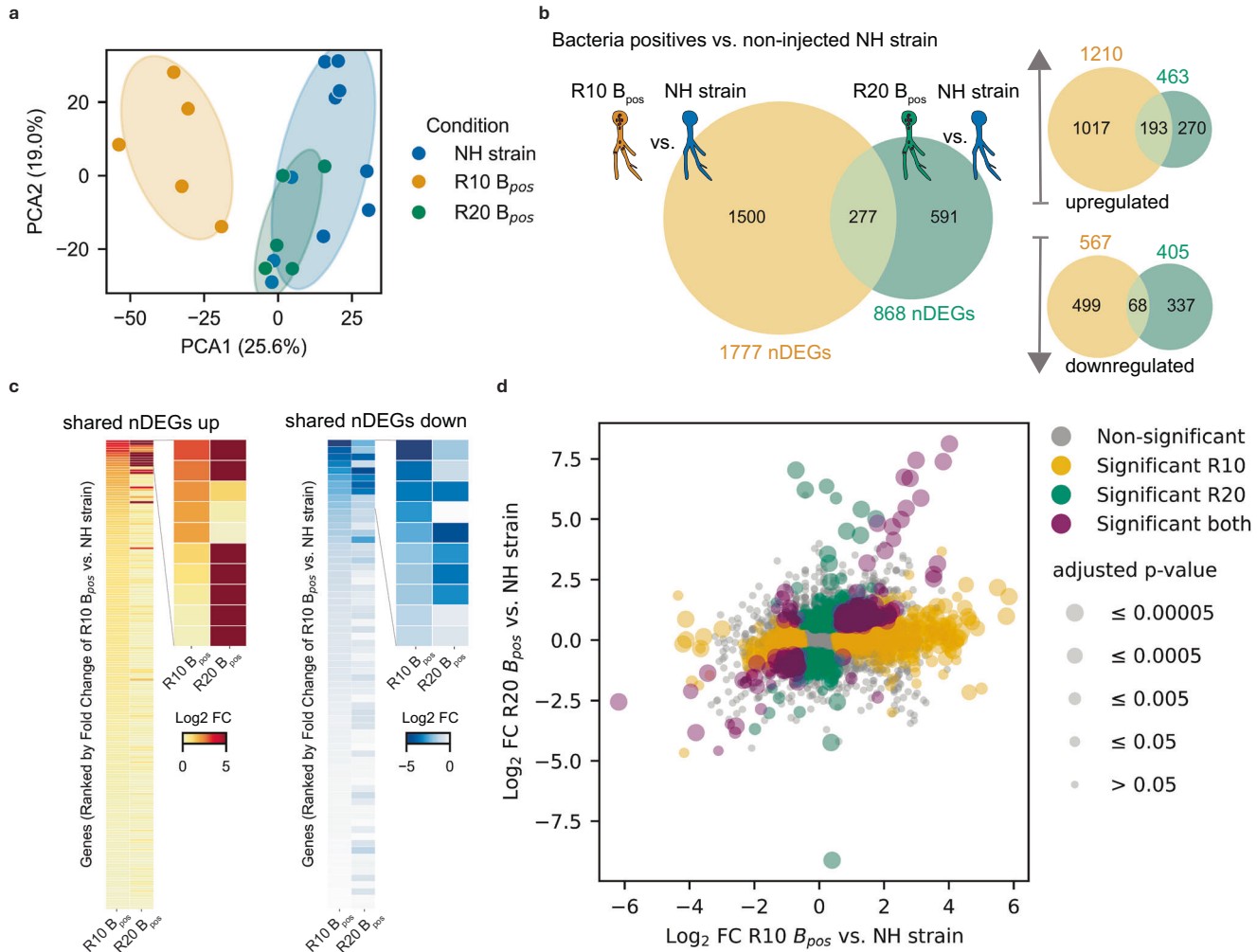

**Fig. 4 | Temporal dynamics of fungal transcriptional responses during prolonged cell-in-cell interaction. a** Principal component analysis of gene count data from RNA sequencing shows that the global expression profile in early round 10 is more distinct from the non-injected NH-strain than the late round 20. **b** Number of differentially expressed genes (nDEGs) of comparison of round 10 versus non-injected NH-strain (R10 B+, orange) and round 20 versus non-injected NH-strain (R20 B+, green). **c** Displayed are the Log2 Fold Changes (FC) of the 196 shared upregulated and 72 downregulated DEGs (left part of each heatmap) with a close view on the top 10 genes in each group (right part of each), most shared DEGs

between the R10 B+ and R 20 B+ are up or downregulated to similar degree in both states; differentially expressed genes are ranked according to their fold change in R10. **d** Combined expression analysis of R10 B+ and R20 B+ versus the NH-strain for identification of common and unique DEGs (two-sided Wald test with Benjamini–Hochberg correction for multiple comparisons with significance cut-off values: Log2 Fold Change > 0.6 ( - 1.5 absolute fold change) and adjusted $p$ value < 0.05, gray marked genes are below the significance threshold). Source data are provided as a Source Data file.

## Discussion

The experimental induction of an intracellular association between a free-living strain of *R. pickettii* and an endosymbiont-free *R. microsporus* provides a tractable model for endosymbiogenesis. By following the trajectory from initial pathogenesis to emerging commensalism, our approach offers insights into scenarios of symbiotic integration. The injection of bacteria via FluidFM technology into fungal cells mimics a cytosolic entry through natural processes such as wounding or hyphal rupture and allowed us to study both bacterial fate and host responses during this novel encounter. Such mix-and-match situations are likely at the origin of emerging endosymbioses in nature but are intrinsically challenging to study in established endosymbiotic model systems[2,14,40].

The ability of *R. pickettii* to proliferate robustly within the fungal mycelium following injection, despite lacking a history of co-evolution with the host, highlights its potential for opportunistic intracellular persistence. Notably, the observed intracellular proliferation dynamics differ from those during the interaction between the natural

endosymbiont *M. rhizoxinica* and its fungal host, indicating that *R. pickettii* employs a distinct, more prolific colonization strategy in this novel association[30]. In mature mycelium, *R. pickettii* surpasses the colonization levels of the native endosymbiont *M. rhizoxinica* by up to two orders of magnitude in both NH and EH strains (Supplementary Fig. 7). This increase suggests that *R. pickettii* exhibits a pathogen-like lifestyle, resembling facultative pathogens that hyperproliferate in the host cytoplasm[41,42]. The pathogenic responses observed in this work may reflect the critical initial barriers that established endosymbiotic systems encountered and overcame during their evolutionary origins, particularly given the fungal host's intrinsic protective mechanisms against bacterial invasion[35]. In addition, the reduced fitness of bacteria-negative spores (Supplementary Fig. 5c) suggests that parental pathogen exposure can cause systemic changes in progeny, leading to fitness costs without direct challenge[43,44]. Pressure-mediated redistribution of bacterial cells within the expanding mycelium appears to create discrete, bacteria-enriched zones (Supplementary Fig. 3). We speculate that these sites may represent microcompartments resulting

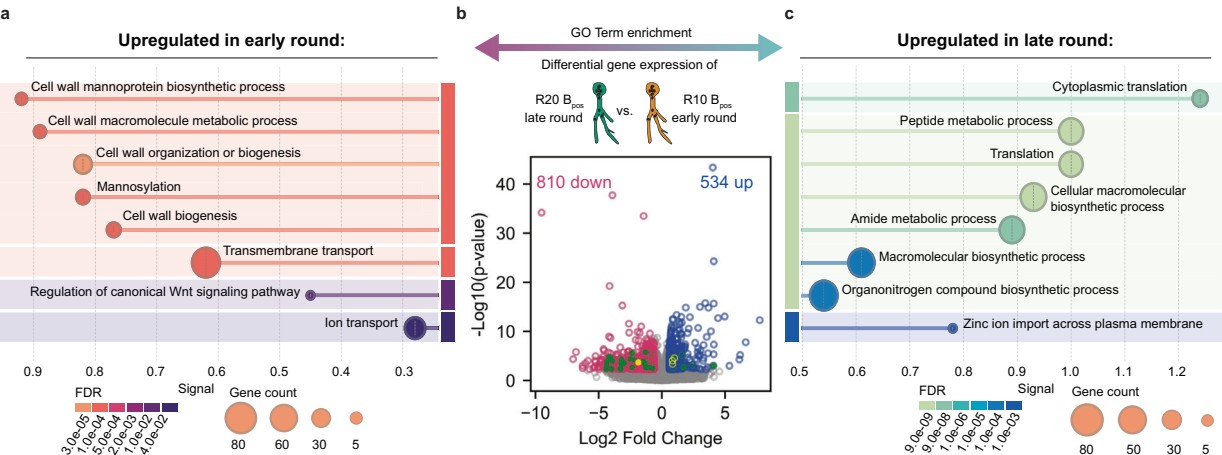

**Fig. 5 | Attenuation of fungal pathogenic transcriptomic responses between serial passages. a** Enrichment analysis by biological process on genes which are significantly upregulated in R10 B$_{pos}$ compared to R20 B$_{pos}$ samples. **b** Differential gene expression analysis between late round (R20 B$_{pos}$) and early round (R10 B$_{pos}$) displaying significantly up- (blue) and downregulated (pink) genes (significance cut-off values: Log2 Fold Change > 0.6 (~1.5 absolute fold change) and $p$ value < 0.05 (adjusted according to Hochberg in DeSeq2), additionally highlighted genes associated with cell wall remodeling (green circles) and reactive oxygen species (yellow circles), full circles show differentially expressed genes that indicate a defense response described by ref. 5 while empty circles follow a different pattern (see Supplementary Data 2) and gray marked genes are below the significance threshold. **c** Enrichment analysis by biological process on genes which are significantly upregulated in R20 B+ compared to R10 B+ samples. **a, c** String analysis with up- and downregulated genes was performed against the uploaded NH proteome (STRG0A18OCV) and biological process terms were grouped by a similarity factor of 0.6 and ranked by their signal (i.e., weighted harmonic mean of enrichment ratio (between observed and expected occurrence) and significance level (LogFalse Discovery Rate (FDR)). Source data are provided as a Source Data file.

from host-mediated functional segregation, similar to pathogen-containing vacuoles in other eukaryotic systems[4,45,46]. The occasionally observed apparent encapsulation of bacteria in vacuole-like structures (Supplementary Video 4) might reflect an adaptive compromise between bacterial sequestration while still allowing for their vertical transfer.

Throughout the serial passaging under low bacterial load regime (B+), the fungal host increased its tolerance to the intracellular bacteria, manifested through an enhanced germination efficiency of B$_{pos}$ spores and improved endosymbiont retention following nutrient deprivation (Fig. 2). This trajectory appears driven by a shift from active defense to tolerance rather than elimination of bacterial antagonistic traits, a transition evidenced in our transcriptomic data showing temporal dynamics in fungal defense gene expression (ref. 5, Supplementary Data 2). Interestingly, despite increased resilience against bacterial burden and improved germination capacity, late-stage adapted spores exhibited a reduced growth rate (Fig. 3), reflecting fitness tradeoffs documented in another fungal-bacterial system consisting of *Mortierella elongata* and its endobacterium *Mycoavidus cysteinexigens*[47,48].

Our temporal classification distinguishes early-stage (round 10) and late-stage (round 20) passages based on distinct fitness profiles (Fig. 2b, c). The transition towards a more resilient, homeostatic expression profile occurred along two primary axes of fungal defense response: remodeling of cell wall synthesis and recalibration of ROS-associated gene expression.

Genes involved in fungal immunity were predominantly upregulated during the early stage, both in comparison to the non-injected NH strain and to late-stage samples (see Figs. 4, 5 and Supplementary Data 2). Chitin synthases, typically induced by general cell wall stress and known to mediate mutualistic interactions between the natural endosymbiont *M. rhizoxinica* and its host in pre- and physical contact assays[5,49,50], were downregulated in early-stage samples. These expression profiles suggest a divergent response from that observed in natural mutualism. In contrast, the early-stage upregulation of chitin-modifying genes; mannosyl transferases and septins, mirrored expression profiles associated with antagonistic interactions between nonhost strains of *R. microsporus* with bacteria[5,51]. ROS-balancing gene expression exhibited more complex dynamics: three out of five

homologs associated with defense-related ROS pathways were upregulated at the late stage, while two displayed the expected early-stage induction characteristic of an antagonistic response (Fig. 5b, yellow circles and Supplementary Data 2). Natural isolates of *R. pickettii* promote iron availability through siderophore production under free-living conditions, which may influence ROS dynamics of associated community partners[52–54]. The identified biosynthetic gene cluster for rhizoferrin synthetase may contribute to these changes in iron homeostasis and ROS production in this novel association[37–39].

From a broader perspective, our findings support the hypothesis that facultative pathogens can serve as intermediates toward stable cell-in-cell interactions under appropriate selective constraints, due to their intracellular survival strategies[55–58]. Our experimental data align with modeling predictions which suggest that endosymbioses may originate from antagonistic interactions and stabilize through repeated rounds of vertical transmission[13,14]. Unlike the rapid bacterial load reduction and full fitness recovery with *M. rhizoxinica*[30], the *R. pickettii* association retained high loads with partial host fitness recovery, suggesting adaptive resilience mediated by attenuated fungal defense. Our work establishes an experimentally tractable model for investigating the principles underlying endosymbiotic integration, showing that pathogenic traits can be redirected toward cooperative persistence. The observed switch between antagonism and commensalism reveals the adaptive capacity of the fungal host to accommodate novel endosymbionts, providing empirical support for pathogen-to-mutualist transition.

## Methods
### Strains and standard culturing methods
The strains and chemicals used are listed in Supplementary Data 3. The fungal strains (*R. microsporus* strain NH is *R. microsporus* CBS 631.82; *R. microsporus* strain EH is *R. microsporus* ATCC62417) were cultivated as previously reported[59]. *R. pickettii* K-288 was cultured routinely in Nutrient-Broth medium at 28 °C. *R. pickettii* K-288 was fluorescently labeled with mCherry by cloning the mCherry coding sequence into pBBR-P12-GFP (ref. 21). *R. microsporus* was grown on potato dextrose agar (PDA) (ThermoFisher) at 28 °C with $20\,\mu g\,mL^{-1}$ gentamicin to prevent extracellular growth of bacteria. For bacterial plasmid retention, addition of $34\,\mu g\,mL^{-1}$ chloramphenicol was used for the culture

of bacteria in all experiments, including the growth of bacteria inside the fungus.

### Re-infection assay

The re-infection assay was performed as previously described[27] with slight modifications. In brief, a 5 mm mycelial disk obtained with a biopsy punch was placed centrally in each well of a 24-well flat bottom plate filled with 750 μL MGYM9 medium without antibiotics and the plate was incubated at 28 °C in the dark. To minimize cross-contamination, samples were spaced apart or grown on separate plates. After 24 h, 100 μL of growing bacteria (OD600 = 1) were added to each well; wells without bacterial inoculation served as negative controls. Spores were harvested on day 12 and were enumerated and analyzed by flow cytometry and microscopy.

### Injection of *Ralstonia pickettii*

The experimental procedures closely followed previously established protocols for instrument setup, FluidFM probe processing, cleaning, and coating[30,60]. Fungal samples were prepared by seeding spores into 50 mm WillCo glass bottom dishes containing 4 mL potato dextrose broth (PDB) medium with 34 μg mL$^{-1}$ chloramphenicol. After incubation at room temperature for 14–16 h, the medium was replaced with a protoplasting solution (1.6 g cellulase Onozuka R10, 40 mg chitinase in 40 mL MMB buffer, pH 5.5) to partially soften the cell wall. Samples were incubated for an additional 3–5 h, with filtering through a 0.22-μm syringe filter if too dense. *R. pickettii* was grown overnight in 10 mL shake flasks at 28 °C, washed three times in Hepes2 buffer (10 mM HEPES, 150 mM NaCl, pH 7.4) and adjusted to an optical density of 2 at 600 nm. An 18-μL bacterial suspension was loaded into the FluidFM probe reservoir. The probe was positioned at the glass surface (z = 0 μm) near the target, retracted to a z position of +8 to +10 μm, and advanced 10 μm for puncturing. Backflow was prevented by pressurizing the microfluidic system to 3.5 bar. To induce influx of bacteria, the microfluidic system was set to a pressure of 6.5 bar, followed by immediate reduction to 3–4 bar to prevent bursting of the germling, after a sufficient volume was injected. Injection success was verified by fluorescence microscopy, and the pressure was gradually reduced to 0 mbar to allow for recovery. The injected germlings were transferred to recovery medium (3.8 mL MMB, 1 mL PDB, 160 μL 4 M sorbitol) within 3–10 min or when growth resumed. Recovery and subsequent dynamics were followed by time-lapse imaging in brightfield and fluorescence modes. Germlings were grown overnight in recovery medium, detached from the probe using overpressure and mechanical assistance, and transferred to potato dextrose agar plates for further incubation at 28 °C. Unless mentioned otherwise, imaging was performed on an inverted AxioObserver microscope equipped with a spinning disk confocal unit (Visitron, Germany) with a Yokogawa (Japan) CSU-W1 scan head and an EMCCD camera system (Andor, UK)[61]. Image acquisition was controlled using the VisiView software (Visitron). Linear adjustments and video editing were made with Fiji.

### Harvesting of spores

Spores were collected 6 ± 1 days after injection or plating from spores, unless mentioned otherwise. Between 12 and 16 mL of spore solution (8.5% NaCl, 1% Tween 20) was added to plates before detaching the mycelium by scraping with a spatula. The soaked mycelium was squeezed using the spatula to release the remaining spore suspension. The spore solution was filtered through a 10-μm CellTrics filter (Sysmex). The spores were washed three times with 5 mL Hepes2 (1000 g; 2 min) and stored at 4 °C until FACS, or in 50% glycerol at −20 °C (for working stocks) or −80 °C (for long-term storage).

### Flow cytometry and cell sorting

Spore analysis and sorting were performed at the ETH Flow Cytometry Core Facility using a FACSAria Fusion BSL2 cell sorter (BD). Individual spores were identified by SSC-A vs. FSC-A and FSC-H vs. FSC-A gating. Bacterial colonization was assessed using SSC-A vs. mCherry-A gating (see Supplementary Fig. 11 for gating strategy). In cases where autofluorescence was suspected to interfere with gate positioning, the mCherry-A vs. PerCP-Cy5-5-A channel was examined to identify autofluorescence; however, this channel was not used for gating. To determine the proportion of positive spores, 100,000 to 1,000,000 spores were analyzed, depending on the sample size. For bulk sorting, spores were collected in 1.5 mL screw cap tubes (Sarstedt), while individual spores were sorted into 96-well plates containing 125 μL PDB supplemented with 34 μg mL$^{-1}$ chloramphenicol and 10 μg mL$^{-1}$ gentamycin per well to assess germination success. Microscopic checks were periodically performed on bulk-sorted spores to validate positive gates. Data analysis was performed using FlowJo v10 software (BD).

### Serial passaging experiment

Spores from three individual injections after the first round were collected and sorted to assess germination success. Each injection was done on a single germling from the same stock, which after recovery, cultivation on plates and sorting gave rise to a population of spores partially colonized by *R. pickettii* (i.e., round 1). The positive populations (B$_{pos}$) were subdivided into high (B++) and low (B+) bacterial load in each round. These lines were maintained individually throughout the experiment (6 lines in total). Four of these lines were maintained for five and two lines for 20 consecutive rounds. For each line in every round, the germination success was measured by optical observation, and the proportion of bacteria-positive spores was measured by FACS. Bacterial spore colonization remained below the levels reported for adaptation of the NH strain with *M. rhizoxinica* (up to 28%, ref. [30]), representing a technical limitation that prevented the parallel maintenance of multiple lines during passaging. For determination of the germination success, single-spore sorted 96-well plates with 125 μL PDB (34 μg mL$^{-1}$ chloramphenicol, 10 μg mL$^{-1}$ gentamycin) per well were incubated at 28 °C and visually inspected for germination using a Zeiss SteREO Discovery.V8 microscope (Zeiss). For each sample, three plates with positive spores and one plate with negative spores were sorted unless mentioned otherwise. Germlings were counted 2 days after sorting. For positive plates, at least five germlings were checked microscopically on day 2 to confirm the presence of fluorescent bacteria. The product of germination success and positive fraction was calculated as fitness index to describe the fraction of viable offspring harboring bacteria. In each round, 100,000 bulk-sorted spores were used for the start of the consecutive new round. Exceptions had to be made for the following reasons: First, due to the low bacterial abundance in spores after injection 2 (0.009%), only 3000 bacteria-positive spores could be collected for that line in round 1. Second, due to logistical constraints during FACS sessions (injection 3, round 4, B++; and rounds 15, B+ and B++), 50,000 spores were collected to start the next round. In each round, 120 μL of bulk-sorted spore solution was spread in five parallel lines on 120 × 120 mm square Petri dishes (Greiner) filled with PDA supplemented with 10 μg mL$^{-1}$ gentamycin and 34 μg mL$^{-1}$ chloramphenicol.

### Bacterial load determination by quantitative PCR

Cryo-preserved spores from B+ round 19 and EH x Mr GFP (natural host carrying GFP labeled *M. rhizoxinica*) as well as the evolved NH x Mr strain (NH strain evolved with *M. rhizoxinica*) from our previous study[30] were harvested as described in the section "Harvesting of spores". Single spores from the full positive population (B$_{pos}$; mCherry for *R. pickettii* in NH, GFP for *M. rhizoxinica* in EH and NH) were sorted directly onto PDA plates supplemented with 10 μg mL$^{-1}$ gentamycin and 34 μg mL$^{-1}$ chloramphenicol. The samples were cultivated at 28 °C until a mycelial diameter of around 30 mm was reached.

Fungal and bacterial genomic DNA was purified from mycelial samples using the DNeasy Plant Pro kit (QIAGEN) with slight modifications. Briefly, mycelium grown on agar plates was transferred to ZR BashingBead lysis tubes (Zymo-Research) and frozen in liquid N₂. Following addition of lysis buffer, samples were incubated at 65 °C for 10 min, before being disrupted twice (3 min each) at 30 Hz in a TissueLyser with a 5 min cooling step on ice between runs. Eluted DNA was quantified using the QuantiFluor ONE dsDNA system (Promega).

Relative quantification of the endobacterial populations was carried out by quantitative PCR (qPCR). Briefly, primer pairs targeting different housekeeping genes were designed: the actin homolog *ACT1* for *R. microsporus* (5′-GTGATGAAGCCCAATCCAAGAG-3′, 5′-TATA-GAAGGTGTGGTGCCAGAT-3′), RNA-polymerase β-subunit *rpoB* for *M. rhizoxinica* (5′-GCTGGATTTGAACGACCAGTT-3′, 5′-GGTCCAGGT-CATCCAGGTATT-3′) and alkyl hydroperoxide reductase *ahpC* for *R. pickettii* (5′-GCTGATCAATACTCAAGTCCAACC −3′, 5′-AGATCAG-CACGGACCACTT-3′). The binding and amplification specificity of each primer pair was verified by conventional PCR and qPCR (bacterial loci only). Template plasmids containing the target DNA sequences were constructed and measured at six different concentrations, ranging from $10^{-2}$ to $10^{-7}$ ng μL⁻¹ to test for amplification efficiency. To determine the bacterial load of biological samples (that is, ratio of fungal to bacterial copy numbers), synthetic samples with a known copy number ratio of 10:1 were used as a reference (as described in ref. [62]). All qPCR reactions were carried out using the FastStart SYBR Green Master Mix (Rox) (Sigma-Aldrich) in a final volume of 25 μL.

### Determination of germination success and bacterial clearance after nutrient starvation

Cryo-preserved spores (50% glycerol, −80 °C) from round 2 and 20 originating from the B+ line were passaged once on PDA plates supplemented with 10 μg mL⁻¹ gentamycin and 34 μg mL⁻¹ chloramphenicol. Single spores from the full positive population ($B_{pos}$) were sorted into 96-well plates filled with 75 μL of Hepes2 (supplemented with 34 μg mL⁻¹ chloramphenicol and 20 μg mL⁻¹ gentamicin). For round 2 at least 192 and for round 20 at least 288 single spores were sorted and analyzed (that is at least two or three 96-well plates per time point). For each sample, negative and positive spores were sorted and analyzed. Plates were then incubated at 16 °C, followed by activation by adding 125 μL of PDB at the respective day of storage and incubated for two days at 28 °C. Germinating spores were counted as described in the section "Flow cytometry and cell sorting". The presence or absence of bacteria after nutrient starvation and activation was verified by fluorescence microscopy in 10x to 40x magnification (inverted AxioObserver (Zeiss) microscope) directly after counting of germinated spores.

### High-throughput imaging

For the imaging experiment, cryo-preserved spores from round 2, 11 and 20 originating from the B+ line were thawed on ice, bulk sorted for positives (entire population $B_{pos}$) and plated on PDA plates supplemented with 10 μg mL⁻¹ gentamycin and 34 μg mL⁻¹ chloramphenicol. Spores were harvested and bulk sorted again to generate fresh inoculum for imaging. For the imaging setup, ultrapure agar (Difco, Europe) was mixed with sterile filtered PDA before autoclavation to prevent impurities. Petri dishes were filled with 20 mL of molten PDA agar supplemented with gentamicin and chloramphenicol. After solidification, 5 × 5 mm spots were cut out off the agar and turned around to expose the even surface. From the bulk sorted spores, 5 μL (or 5000 spores) were spotted onto the agar surface, quickly dried and transferred top-to-bottom into 8-well chambers for imaging (Nunc™ Lab-Tek™ II, Thermo Europe). To facilitate timelapse imaging, the agar-pads were fixed with additional 250 μL agar in the 8-well chamber. Imaging was performed on an inverse Nikon Eclipse Ti2 microscope equipped with a spinning disk unit (Yokogawa CSU-W1-T2) and a Hamamatsu Orca Fusion BT camera. Timelapse imaging was done up to 16 h with a time interval of 15 min and over z-length of 16 μm (1 μm per slice). Illumination was done using a Lumencor SpectraX Chroma laser and emission filter (GFP 525/50 & CY3-RFP 600/52) in 100x magnification (100 × 1.45 CFI Plan Apo Oil objective). Typically, 40 positions were imaged per timeframe.

To further understand how bacteria influence spore germination, we developed an automated Python-based pipeline to segment and track spores during their early germination stages. We trained three distinct deep-learning models using a U-Net architecture with a resnet34 encoder, implemented through the Segmentation Models library built on Keras and TensorFlow (https://github.com/qubvel/segmentation_models).

The first two models were trained on fully annotated white-field images (68 images with 1414 spores) of spores to detect entire cells and their contours, respectively. The third model was trained on the same dataset, but with annotations (217 images with 3941 spores) limited to the spore body, excluding germ tubes. Predictions from all three models were then integrated, using the contour model to separate touching spores, and the spore body model to confirm that segmented objects were connected to a cell body. To minimize tracking errors at later timepoints, when germ tubes become prominent across the field of view, we also monitored the roundness of segmented objects using formula

$$Roundness = \frac{4\pi * area}{perimeter^2} \qquad (1)$$

A minimum roundness threshold 0.6 was applied to filter out elongated objects, ensuring that only early germinating spores were included in the analysis.

Spore tracking was then performed by matching segmented objects across consecutive frames based on spatial overlap of the maximal modal value. For each tracked spore, we quantified the following parameters over time: (1) area, representing the segmented object's surface; (2) length, defined as the cumulative length of the skeletonized object; (3) roundness, as described above; and (4) cumulative fluorescence intensity, calculated from the sum projection of the z-stack of the mCherry channel within each segmented object. All codes and annotated images are available on GitHub (https://github.com/BDehapiot/ETH-ScopeM_Gassler, ref. [63]).

### Transcriptomics

Samples for transcriptome analysis were obtained from three conditions: non-host wild type (NH-strain), as well as round 10 and round 20 isolates of the low bacterial load line (B+). Thawed cryostocks were maintained at 4 °C and sorted by flow cytometry using mCherry-based gating (Supplementary Fig. 11). At least 100,000 spores were collected per gate.

For round 10 and round 20 samples, mCherry-positive spores ($B_{pos}$) and, for the NH-strain, mCherry-negative spores ($B_{neg}$) were plated on PDA supplemented with 10 μg mL⁻¹ gentamycin and 34 μg mL⁻¹ chloramphenicol, followed by 6 days of incubation at 28 °C. Freshly harvested spores were then re-sorted into $B_{pos}$ (round 10, round 20) or $B_{neg}$ (NH-strain, round 10, round 20) fractions into 96-well plates containing 125 μL PDA medium supplemented as above (one spore per well). Cultivation proceeded for 24 h at 28 °C. Wells designated $B_{pos}$ were microscopically checked for bacterial presence prior to harvest.

Biomass from each well was collected using tweezers, plunge-frozen in liquid nitrogen, and transferred to tubes containing polystyrene beads (ZR BashingBead™ 2.0 mm, Lucerna Chem AG). Five biological replicates were prepared per condition, with each replicate comprising a pooled biomass from ten individually grown germlings.

Samples were kept on dry ice during collection and stored at −80 °C until processing.

For RNA extraction, 1 mL of preheated (65 °C) QIAzol lysis reagent (Qiagen AG) was added to each frozen sample. After 1 min at room temperature (RT), samples were vortexed for 3 min, incubated for an additional 2 min at RT, and then disrupted twice (3 min each) at 30 Hz in a TissueLyser with a 5 min cooling step on ice between runs. Subsequently, 200 μL chloroform was added, tubes were thoroughly mixed, and samples were incubated at RT for 3 min. Phase separation was achieved by centrifugation (12,000 g, 15 min, 4 °C). The aqueous phase was transferred to a new RNase-free tube and mixed 1:1 (v/v) with 100% ethanol before purification using the RNeasy Kit (Qiagen) according to the manufacturer's instructions. RNA was eluted in 30 μL RNase-free water, quantified by NanoDrop, and adjusted to 200 ng μL⁻¹. Typical DNA:RNA ratios ranged from 20:80. Residual genomic DNA was removed using the TURBO DNA-free kit (Thermo, Europe) in a 50 μL reaction as per the manufacturer's protocol. DNA-free RNA was stored at −80 °C in two aliquots: one for quality control and one for library preparation to avoid multiple freeze-thaw cycles. Quality control comprised quantification of residual DNA (QuantiFluor ONE dsDNA, Promega) and RNA (QuantiFluor RNA, Promega), and assessment of RNA integrity with a Fragment Analyzer (Agilent). Samples with <5 ng μL⁻¹ residual DNA and an RNA Integrity Number (RIN) > 6.5 were used for subsequent library preparation and sequencing at BMKgene (Germany).

Poly-A enriched libraries were prepared and sequenced by BMK Gene (https://www.bmkgene.com/) on an NovaSeq 6000(Illumina) using paired-end 150 bp sequencing. The resulting raw reads were cleaned by removing adapter sequences, low-quality-end trimming, and removal of low-quality reads using BBTools v 38.18 [Bushnell, B. BBMap. Available from: (https://sourceforge.net/projects/bbmap/). The exact commands used for quality control can be found on the Methods in Microbiomics webpage (Sunagawa, S. Data Preprocessing −Methods in Microbiomics 0.0.1 documentation. Available from (https://doi.org/10.5281/zenodo.15019381, ref. 64). The quality-controlled reads were aligned against the *R. microsporus* assembly from our previous work[30] (GCA_964213375.1) using STAR v. 2.7.8 aligner[65]. Transcript abundances were quantified using featureCounts (ref. 66, subread v. 2.0.1). Differential gene expression analysis was performed using Bioconductor R package DESeq2 (ref. 67, v. 1.37.4). Functional enrichment analysis was performed using StringDB[68] v.12. The *R. microsporus* predicted proteome (obtained by extracting protein sequences from the genome assembly GCA_964213375.1 from our previous work[30]) was uploaded to StringDB v.12.0.2 using the 'Add organism' feature (organism identifier STRG0A18OCV). The set of upregulated and downregulated genes were analysed against STRG0A18OCV predicted proteome for enriched biological processes. A list of all permalinks and fasta files to the respective StringDB entries is provided in Supplementary Data 1 and in the Supplementary Material. All raw RNA sequencing data are available in the European Nucleotide Archive under the accession number PRJEB89230.

### Declaration of generative AI and AI-assisted technologies
The authors used ChatGPT to streamline the human-generated code for plotting data in graphs in Figs. 1–5 and Supplementary Fig. 2, 5, 6, 7, 9 and 10. After using the tool, the authors reviewed and edited the code, and take full responsibility for the content of the publication.

### Reporting summary
Further information on research design is available in the Nature Portfolio Reporting Summary linked to this article.

## Data availability
The RNA sequencing data generated in this study have been deposited in the European Nucleotide Archive under accession code PRJEB89230. The FACS data and raw images used in this study are available in the Zenodo database (ref. 69, https://zenodo.org/records/16846729). Source Data for all relevant Fig. (1f, i and j; 2b–d; 3b–d; 4a–d; 5a–c) and Supplementary Figs. (2a, b; 5b, c; 6; 7g; 9; 10a, b) are provided with this paper. Source data are provided with this paper.

## Code availability
All the original code for the analyses and visualization has been deposited in the Supplementary Material together with all source data and will be made available on GitHub (https://github.com/MicrobiologyETHZ/tgassler_rhizopus, ref. 70). All codes and annotated images for analysis of fungal germination are available on GitHub (https://github.com/BDehapiot/ETH-ScopeM_Gassler, ref. 63).

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

## Acknowledgements

This work was supported by a European Research Council Advanced Grant (SYMBIOSES number 883077 to J.A.V.) and an ETH Zurich Postdoctoral Fellowship from the ETH Research Grant (Number 21-1 FEL-08 to T.G.). We thank E. Sarajlic for technical assistance. We kindly acknowledge the support of the ETH Flow Cytometry Core Facility team (F. Mair, R. Antonialli, A. Schütz, I. Vgenopoulou and M. Kisielow). We thank Markus Kuenzler for helpful discussions during the project.

## Author contributions

Conceptualization: T.G., G.G., and J.A.V.; Data curation: T.G. and A.S.; Formal analysis: T.G., A.S. and B.D.; Funding acquisition: T.G. and J.A.V.; Investigation: T.G., G.G., O.X.B., A.H., and M.B.M.; Methodology: T.G., J.A.V., G.G., S.S., A.S. and B.D.; Software: B.D. and T.G.; Resources: J.A.V and S.S.; Visualization: T.G.; Writing original draft: T.G. and J.A.V.; Review & editing: all authors.

## Funding

## Competing interests

The authors declare no competing interests.
