## [Transparent Peer Review file · Nature Communications]

Induced endosymbiosis between a fungus and bacterium reveals a shift from antagonism to commensalism

Corresponding Author: Professor Julia Vorholt

Version 0:

Reviewer comments:

Reviewer #1

(Remarks to the Author)

This ms is a fascinating follow-up to the group's 2024 Nature paper (Giger et al 2024). In the current study by Gassler et al, the authors use the novel fluid injection method, developed in the Giger et al 2024 paper, to establish a new induced endosymbiosis between the fungus *Rhizopus microsporus* and *Ralstonia pickettii* bacteria, which was subsequently experimentally evolved in the lab and analysed. In this way, the current study established a brand new symbiosis between *Rhizopus* and bacteria, which would otherwise not exist in nature, highlighting the plasticity of fungi to adapting to new bacterial endosymbionts under the right selective pressure. The current study performed detailed analysis of how the fungal-bacterial interaction changed as a result of serial passage through multiple cultivation rounds. The main findings indicate that the newly established symbiosis reduced fungal fitness, but by round 20, fitness recovered, albeit not to WT levels. This was corroborated by transcriptomic analysis of the fungus, which appeared to behave more similarly to the WT endosymbiont-free strain (NH) by round 20 as compared to round 11. Overall, this study created an experimentally tractable and fully manipulable system for studying the establishment of novel fungal-bacterial endosymbioses and thus contributes exciting new knowledge to the fields of fungal biology, microbiology, experimental evolution and microbial interactions.

The manuscript is carefully crafted and well-written. Below are some comments and suggestions for improvement:

As I understand, the reasons Rm NH strain was used for these experiments is due to its ability to sporulate and the need for germlings in the injection experiments. This, however, is not clearly stated in the ms, and instead some readers might wonder why the EH strain wasn't chosen for these experiments. I would suggest mentioning your rationale for choosing the NH strain more clearly.

Throughout the ms, authors interchangeably refer to the different fungal populations which were passed through multiple rounds of cultivation as "rounds", "lines" and even "timepoints". I would suggest to choose one of these terms (round or line would be my suggestion), provide a clear definition of it, and then make sure to maintain consistency of this terminology throughout the ms to avoid confusion.

Line 119-120 Fluorescence microscopy confirmed an increased bacterial load in B++ over B+ spores (Figure 1 h and i). In figure 1h, only a comparison between B- and B++ is shown, it would be good to include here pictures of B+ for comparison too.

Lines 31-33: it is unclear why injection experiments 1&2 were subjected to 5 rounds whereas injection experiment 3 was maintained for 20 rounds. Is it that after 5 rounds Injections 1&2 died? Later, in line 143, it is stated that one line was passed on through 20 rounds to assess dynamics of the interaction, but this could have been done with all three injection experiments, so why was it not done?

Line 139-140: what are the 6 replicates here? Was each injection line performed in duplicate?

Line 145-146: "Concurrently, germination rates of bacteria-negative spores followed a similar trend, indicating a systemic fitness deficit". This is counterintuitive, as one would expect the bacteria-negative spores to have higher fitness. It would be nice to see some discussion on this in the "discussion" section.

Figures

Figure 2. The figure caption needs to be reorganised to clarify what each part of the figure is showing. The way it is currently formatted is hard to understand.

Figure 2b-d. The bar graphs need additional explanation – is each “bar” an average from the three injection experiments? And if yes, there can be no average for “rounds” beyond 5 as only injection 3 was brought to 20 rounds. What do the triangles/squares represent? It is not clear, that the colour-coding is according to figure 2a and represents the different “rounds”, it needs to be explicitly stated with a legend perhaps. What do the dotted black vertical lines represent, and the dotted red line in 2b?

Figure 2e, needs clarification that this is showing response to nutrient starvation/limitation. It is currently not clearly stated in the figure itself, only in the caption (and in a rather convoluted way).

Line 185-186: please clarify whether the 384 individually tracked spores were for each of the “rounds” (R2, 11, 20) or the total number of spores tracked? If the latter, how many spores per round were analysed/tracked?

Line 192: please include the statistical test used to generate the p-value here.

Line 125-216: “timepoints” refers to R10 and R20? I suggest changing to “rounds”

Line 258: typo in word “symbiotic”, change to “symbiotic”

Materials & Methods

Section “strains and standard culturing media” requires details on the antibiotics used for plasmid maintenance. Gentamicin only is mentioned here, but in later sections there is reference to chloramphenicol, please clarify the purposes of these respective antibiotics and their concentrations. Moreover, where was the bacterial strain obtained from and any details on what environment it was originally isolated from?

The section “re-infection assay” requires more explanation. It is not clear what the “wells” on a plate refer to, what kind of “cross contamination” is being minimised, what “critical samples” are, and why “antibiotic-free wells” are serving as negative controls. Was the media used liquid or solid?

“Serial passaging experiments section” also requires further details. Did each population “line”, whether B++ or B+ originate from a single spore or a collection of spores? If so, how many? It is unclear what constitutes a “round”, please clarify. It is not clear how germination rate was measured: for example, how many spores were tested? How was germination monitored? - if this was done by microscopy please detail the set up.

In section “Test against nutrient starvation and bacterial clearance”, what is meant by cryo-preserved? Spores preserved at -20C or -80C? Also, the title of this section refers to “bacterial clearance”, but it is not explained or detailed in the section.

Line 483: please clarify that this *Rhizopus* assembly is available in NCBI. Was it sequenced in the course of the current study? Is this the same fungal strain as used in the current study?

Line 486 – where is the *Rhizopus* proteome coming from?

(Remarks on code availability)

I have checked the provided code, which is openly available. Unfortunately, I do not have the relevant coding experience especially with image analysis to test its reproducibility.

Reviewer #2

(Remarks to the Author)

The research work submitted by Gassler and collaborators utilized fluid force microscopy to introduce a free-living bacterial strain, *Ralstonia pickettii* K-288, into symbiotic and asymbiotic strains of *R. microsporus*, which the authors refer to as EH and NH, respectively.

Authors hypothesized that implantation of this bacterium in the fungal cytosol will initially be perceived as a pathogen by the fungus, but when positive selection is maintained, this interaction will transition to a more stable association.

To test this hypothesis, the authors first labeled *R. pickettii* K-288 with mCherry to determine if this bacterium could colonize EH, cured EH, or NH strains of *R. microsporus*. They also tested if co-infection of *R. pickettii* and *M. rhizoxinica*, the natural bacterial symbiont of *R. microsporus*, was possible by co-cultivating cured EH and the two labeled bacterial strains (mCherry and GFP-labeled, respectively). These experiments showed that *R. pickettii* was unable to colonize EH or NH strains naturally. Colonization was also not possible when co-cultivation with *M. rhizoxinica* was performed in the cured EH fungus. Did the authors notice any differences in the “external” interaction of *R. microsporus* EH and NH towards *R. pickettii*? Can one speak of *R. pickettii* as a pathogen of *R. microsporus*? Why? What happens to the EH and NH fungus in the external presence of *R. pickettii*?

Using fluid force microscopy, authors injected labeled *R. pickettii* into NH *R. microsporus* (3 independent events) and selected spores that harbor either a low (B+) or a high bacterial load (B++) for five and up to 20 rounds of positive selection. The authors analyzed the fitness index, calculated as the product of the germination rate (i.e., germination success or percentage germination) and the fraction of spores that harbor *R. pickettii*. They noticed that this index changes with serial passaging. These experiments do not appear to reach stability, and the fitness index remains relatively low in both scenarios (high and low bacterial load). This data is based only on injection 3, which was the only one followed and analyzed until round 20. Authors should also analyze the outcomes of the two other events to round 20, thereby better supporting their claims. Do the three independent injections lead to similar results in terms of fitness?

Additionally, when authors use positive spores to generate new ones, they redo the sorting. How many fresh spores are classified again as positive after this selection (percentage of the total)? Is this high? Was it different for R2, R11 and R20?

With spores from injection 3, the authors further analyzed that fungal growth decreased when *R. pickettii* colonized (NH vs R20) and that germination success tended to increase from R2 to R20, having spores from R20 a higher bacterial load. This result contrasts with previous research using *M. rhizoxinica* and NH *R. microsporus*, which found that an important adaptation to increase host fitness was to reduce or control bacterial load. A propo: When comparing B+ vs. B++, what is the average of bacterial cells contained per spore in each case?

Later, the authors studied the fungal transcriptomic response to the intracellular interaction with *R. pickettii*, showing that the transcriptional response between R20pos and NH was more similar than between R10pos and NH, suggesting that the fungus adapted its response to the symbiotic bacterium. Additionally, when authors compared the R20pos vs. R10pos, they identified up-regulated genes in the early round versus those in the late round, indicating the transitions likely required for the establishment of endosymbiosis.

One prominent aspect that authors did not consider in their work is the response and changes that occur to the bacterium during the serial passaging experiment. Therefore, the claim that the interaction changed from an antagonistic to a commensal interaction is not yet fully supported. For this, we need to know the changes suffered by the bacterium at both the genomic and/or transcriptomic levels.

In summary, the presented evidence suggests a host-driven physiological plasticity response rather than true evolutionary co-adaptation. Authors do not report heritable mutations in either fungus or bacteria nor are there functional adaptations suggestive of a transition toward stable endosymbioses, such as gene loss or intracellular specialization. In this regard, it is advisable to use terms such as "induced intracellular association" rather than "endosymbiosis" or "co-evolution" unless authors provide bacterial genomic or transcriptomic data to support a deeper adaptive process emerging through serial passaging.

Minor comments:

Abstract

Lines 6-7, "bacteria implantation" replace with "bacterial implantation"

Line 7 compromised host fitness, as evidenced by reduced fungal viability (add a coma)

Line 24 "predispose them for" replace with "predispose them to"

Line 25 "Such adaptations, may provide" (remove the coma)

Line 31 "remains limited" replace with "remain limited"

Results

L85. Cured strain' is mentioned without a definition

Line 125, "designed an serial" replace with "designed a serial"

Line 194, the hyphen in "NH-strain" should be consistent: either use "NH strain" without the hyphen or always use "NH-strain."

Discussion

Line 257 "symbiotic integration" replace with "symbiotic integration"

Lines 275-6 "pathogencontaining" replace with "pathogen-containing"

Methods

Line 419, 100x magnification (000x 1.45 CFI Plan Apo Oil objective). Its 000x or 100x?

Sup_Video_Legends: Supplementary Video 2 label is repeated. The last one should be Video 3. Beautiful videos!

(Remarks on code availability)

I did not have the time to test the code.

I only checked it was available as stated.

Reviewer #3

(Remarks to the Author)

This is a novel approach to reveal how initial negative influence of endofungal bacteria evolves towards more commensalism. Although the described experimental system might have limited time frame, this provides a simplistic and excellent approach to study such evolutionary transition that might exist in nature. The described data are detailed and thorough, I have no major concerns. However, the writing of the manuscript could benefit from slight attention to describing the various strains and treatments. I understand the author aimed for a condensed text, but when I took down the manuscript for few hours, I was unable to follow all abbreviations. Therefore, I suggest revising the text to describe not only the treatments with abbreviations but also add half a sentence on what was compared (e.g. in line 206-207, I am lost what is compared with what).

Also, I am not fully convinced that fungus displays a triggered immune response, but it might be a stress induced reaction by the fungus. I understand immune response as specific reaction to an immunogen performed by specialized cells or protein pathway, but I only see here stress response being induced in the fungus, no specific protein pathway is induced. Thus, I suggest the authors to remove the term immune response, unless they add additional data describing an immune response like pathway (e.g. line 8, 80, 290, 292). The authors nicely avoid this term in the rest of the manuscript, which I agree with.

Finally, the presence of microcompartment coupled with host-mediated functional segregation is very intriguing idea, and such should be easily visible with using high resolution microscopy or using fluorescent membrane stains. Especially comparing the initially introduced bacterial cell localization and those after adaptation would make the story more compelling if localization is indeed altered.

Line 265: I am not fully convinced about “co-evolved compatibility traits”. It should be explained better what is meant under this term and whether this is specifically demonstrated. No specific molecular mechanisms is revealed, so I am not sure what this could refer to.

Line 307: Is it really pathogen if host is not killed, but delayed for growth? Also, the bacterial entrance would not happen without active introduction of those cells. Thus, this is not pathogenesis but possibly a defector. I understand the need of commonly used terms, but the traditional pathogen term might be misinterpreted here.

(Remarks on code availability)

Version 1:

Reviewer comments:

Reviewer #2

(Remarks to the Author)

The authors did a great job in answering the concerns we reviewers had and also in successfully incorporating most of the major recommendations.

I thank authors for investigating in more detail the effects of the external interaction between *R. microsporus* (EH and NH) and *R. pickettii*. and also for comparing the interactions with *M. rhizoxinica* and *E. coli*. I agree that the focus of the manuscript is the intracellular interaction of *R. microsporus* NH, a naturally asymbiotic fungus, with *R. pickettii*, which is not a natural symbiont of *R. microsporus*. Thus, the generated information is valuable, but not needed to understand the main story of this work.

I also highly value the efforts to estimate the abundance of the bacterial symbionts in the fungal host using qPCR. From this data, it is pretty clear that *R. pickettii* cell numbers within fungal NH mycelia are much higher than those achieved by *M. rhizoxinica*. A propo supplementary Figure 7 it says “(g), bacterial colonization obtained by qPCR shows that *R. pickettii* (recovered round 19 Bpos spores) reaches larger colonization levels than *M. rhizoxinica* in the EH and EH strain”, and it should be “in the EH and NH strain”. A small typo.

Some other changes I still suggest are:

Change the label “Germination rate” for “Germination success” or “Germination” as in your previous report (Giger et al., 2024). This change is required in figures and text along the whole manuscript (for example in Figures 1j, 2b, 2e, 3c, S5, S6, etc). This change will increase the consistency and accuracy of the work, as authors report the percent of spores that did germinate from the total in each round, and not the rate of germination (spores germinated by a unit of time (min, hr, day)).

In Fig. 2f the label for the y-axis should be “Positive fraction (%)” instead of “Positive rate(%)”, right?

I wonder why authors use different scales for Figure 2b (0, 20, 40, 60,..) and 2e (0, 1, 10, 100) if they show germination (%) in both cases. I think it might be easier for the reader to compare numbers if the same scales are use in both panels.

Finally, I am deeply curious about the potential changes that might have occurred to the bacterial genome after 20 rounds of positive selection, and under which circumstances (ecological context) this novel symbiosis would further stabilize. Great work that opens many questions and new possibilities of testing them!

(Remarks on code availability)

Only that it is publicly available, but I did not have the time to test if it works.

Reviewer #3

(Remarks to the Author)

Thank you for adjusting the manuscript according to the suggestions, I have no further comments. Congratulations on the superb work!

(Remarks on code availability)

REVIEWER COMMENTS

Reviewer #1 (Remarks to the Author):

This ms is a fascinating follow-up to the group's 2024 Nature paper (Giger et al 2024). In the current study by Gassler et al, the authors use the novel fluid injection method, developed in the Giger et al 2024 paper, to establish a new induced endosymbiosis between the fungus *Rhizopus microsporus* and *Ralstonia pickettii* bacteria, which was subsequently experimentally evolved in the lab and analysed. In this way, the current study established a brand new symbiosis between *Rhizopus* and bacteria, which would otherwise not exist in nature, highlighting the plasticity of fungi to adapting to new bacterial endosymbionts under the right selective pressure. The current study performed detailed analysis of how the fungal-bacterial interaction changed as a result of serial passage through multiple cultivation rounds. The main findings indicate that the newly established symbiosis reduced fungal fitness, but by round 20, fitness recovered, albeit not to WT levels. This was corroborated by transcriptomic analysis of the fungus, which appeared to behave more similarly to the WT endosymbiont-free strain (NH) by round 20 as compared to round 11. Overall, this study created an experimentally tractable and fully manipulable system for studying the establishment of novel fungal-bacterial endosymbioses and thus contributes exciting new knowledge to the fields of fungal biology, microbiology, experimental evolution and microbial interactions.

We thank the reviewer for their positive assessment of our study and for highlighting the key points of the work.

The manuscript is carefully crafted and well-written. Below are some comments and suggestions for improvement:

As I understand, the reasons Rm NH strain was used for these experiments is due to its ability to sporulate and the need for germlings in the injection experiments. This, however, is not clearly stated in the ms, and instead some readers might wonder why the EH strain wasn't chosen for these experiments. I would suggest mentioning your rationale for choosing the NH strain more clearly.

*Thank you for pointing this out. We have now revised the manuscript to clearly state our rationale for selecting the NH strain over the EH strain for injection experiments. As the reviewer correctly notes, only the NH strain can sporulate in the absence of *M. rhizoxinica*, which is essential for obtaining germlings and ensuring tractable propagation and selection following injection. In addition, our experimental design deliberately pairs a non-host with a non-endosymbiont to establish a novel intracellular interaction. This approach allows us to investigate how compatibility and fitness relationships emerge from a non-symbiotic starting point.*

The changes have been made to lines 63-66.

Throughout the ms, authors interchangeably refer to the different fungal populations which were passed through multiple rounds of cultivation as "rounds", "lines" and even "timepoints". I would suggest to choose one of these terms (round or line would be my suggestion), provide a clear definition of it, and then make sure to maintain consistency of this terminology throughout the ms to avoid confusion.

We have standardized terminology throughout the manuscript. We now consistently use "rounds" to refer to each sequential passage.

We have made the respective changes throughout the manuscript (for example in line 225 and 256).

Line 119-120 Fluorescence microscopy confirmed an increased bacterial load in B++ over B+ spores (Figure 1 h and i). In figure 1h, only a comparison between B- and B++ is shown, it would be good to include here pictures of B+ for comparison too.

Thank you for this comment. We have now included images of all three subpopulations in Figure 1 h to provide a complete visual comparison. In addition, we have added Supplementary Figure 4 with exemplary images that demonstrate the visual differences in bacterial load between B+ and B++ spores.

Lines 31-33: it is unclear why injection experiments 1&2 were subjected to 5 rounds whereas injection experiment 3 was maintained for 20 rounds. Is it that after 5 rounds Injections 1&2 died? Later, in line 143, it is stated that one line was passed on through 20 rounds to assess dynamics of the interaction, but this could have been done with all three injection experiments, so why was it not done?

We conducted three independent injections, and the resulting fungal populations were then subjected to both B+ and B++ passaging regimes for five consecutive rounds, observing a consistent fitness trend of bacterial-positive spores. We then extended one representative injection line to 20 rounds to characterize the long-term interaction dynamics. However, we did not extend all three initial injections to 20 rounds due to practical constraints. The low abundance of bacteria-positive spores in each passage round requires FACS sorting at a core facility, which became rate-limiting in our experimental workflow. This constraint prevented us from maintaining multiple lines in parallel through extended passaging. We have now clarified this experimental design rationale in the revised manuscript (see lines 406-408).

Line 139-140: what are the 6 replicates here? Was each injection line performed in duplicate?

This refers to three independent injections, each of which was subsequently passaged under both high bacterial load (B++) and low bacterial load (B+) conditions, resulting in six experimental lines total between rounds 1 and 5. We have adjusted the text to describe this experimental design more clearly (see line 148 and lines 431-450).

Line 145-146: “Concurrently, germination rates of bacteria-negative spores followed a similar trend, indicating a systemic fitness deficit”. This is counterintuitive, as one would expect the bacteria-negative spores to have higher fitness. It would be nice to see some discussion on this in the “discussion” section.

We agree the observation appears counterintuitive at first, we have added a paragraph to the Discussion section addressing these systemic effects. We hypothesize that the reduced germination rates observed in bacteria-negative spores reflect transgenerational fitness costs that persist even in the absence of direct bacterial colonization in the progeny. Such effects could arise from several mechanisms: pathogen-induced physiological stress in the parent generation, epigenetic modifications transmitted during reproduction, or resource reallocation during sporulation that compromises offspring performance. Similar transgenerational effects following pathogen challenge have been documented in plants, where exposure in one generation influences immunity and fitness in the next one^{1,2}.

We have incorporated this discussion (see lines 304 to 306).

Figures

Figure 2. The figure caption needs to be reorganised to clarify what each part of the figure is showing. The way it is currently formatted is hard to understand.

Figure 2b-d. The bar graphs need additional explanation – is each “bar” an average from the three injection experiments? And if yes, there can be no average for “rounds” beyond 5 as only injection 3 was brought to 20 rounds. What do the triangles/squares represent? It is not clear,

that the colour-coding is according to figure 2a and represents the different “rounds”, it needs to be explicitly stated with a legend perhaps. What do the dotted black vertical lines represent, and the dotted red line in 2b?

Figure 2e, needs clarification that this is showing response to nutrient starvation/limitation. It is currently not clearly stated in the figure itself, only in the caption (and in a rather convoluted way).

We have carefully restructured the legend to clearly describe each panel, explain bar components, color coding, symbols and reference lines. We have also adapted the subpanel 2 e and 2 f to highlight that effects on germination rate and bacterial clearance respectively are shown.

Line 185-186: please clarify whether the 384 individually tracked spores were for each of the “rounds” (R2, 11, 20) or the total number of spores tracked? If the latter, how many spores per round were analysed/tracked?

This represents the total number of spores tracked across all conditions. The specific numbers of tracked / analyzed spores were 151/131 for the NH strain, 49/16 for R2 B+, 92/76 for R11 B+ and 92/62 for R20 B+. We have included these sample sizes in the revised legend of Figure 3.

Line 192: please include the statistical test used to generate the p-value here.

Statistical differences in fungal doubling times across rounds were assessed using the Kruskal–Wallis test, followed by Dunn’s post-hoc test with Bonferroni correction. We have added the explanation in the text and Figure legend.

Line 125-216: “timepoints” refers to R10 and R20? I suggest changing to “rounds”

This is correct and we have changed to consistent nomenclature throughout using “rounds”.

Line 258: typo in word “symbiotic”, change to “symbiotic”

Thank you.

Materials & Methods

Section “strains and standard culturing media” requires details on the antibiotics used for plasmid maintenance. Gentamicin only is mentioned here, but in later sections there is reference to chloramphenicol, please clarify the purposes of these respective antibiotics and their concentrations.

*We have added an explanation of antibiotic usage to the Methods section. Gentamycin is routinely used in the fungal culture because its amino glycoside structure prevents penetration of the fungal cell wall and membrane, thereby inhibiting extracellular bacterial growth. Chloramphenicol is used for plasmid maintenance in *R. pickettii* to ensure mCherry expression. We have included the rationale and working concentrations for both antibiotics in the revised Methods section.*

Moreover, where was the bacterial strain obtained from and any details on what environment it was originally isolated from?

*We use the type-strain of *R. pickettii* K288, which we ordered from DSMZ (entry 6267). This strain was originally isolated from a clinical sample³. Although isolated from a clinical source, *R. pickettii* strains are ubiquitously occurring in the environment (see Figure R1). The information is provided in the Methods section.*

The section “re-infection assay” requires more explanation. It is not clear what the “wells” on a plate refer to, what kind of “cross contamination” is being minimised, what “critical samples” are,

and why “antibiotic-free wells” are serving as negative controls. Was the media used liquid or solid?

We have revised the manuscript to provide a clearer explanation of the setup and add appropriate references⁴. The assay was performed in 24-well plates using antibiotic-free liquid MGYM9 medium to support the growth of both fungus and bacterium under the tested conditions. Cross-contamination between wells, though rare, could occur through aerial spore transfer between adjacent wells. Such events were monitored, and assays were repeated if contamination was detected.

“Serial passaging experiments section” also requires further details. Did each population “line”,

*Figure R1: Environmental statistics on *R. pickettii* showing the distribution of samples containing this operational taxonomic unit per habitat and sub-habitat; source: https://microbeatlas.org/taxon?taxon_id=90_12;96_51;97_54;98_59;99_921; retrieved 04.07.2025.*

whether B++ or B+ originate from a single spore or a collection of spores? If so, how many? It is unclear what constitutes a “round”, please clarify. It is not clear how germination rate was measured: for example, how many spores were tested? How was germination monitored? - if this was done by microscopy please detail the set up.

*We have re-written the respective method section accordingly (see lines 409 and following). Each injection was performed on a single germling, which after injection produced a population of spores that were partially colonized by *R. pickettii* (designated as Round 1). From these Round 1 spores, we collected 100,000 bacterial-positive spores for injection 1 and 3. Due to the low bacterial abundance in spores from injection 2 Round 1 (0.009%), only 3,000 bacteria-positive spores could be collected for that line. Subsequently, 100,000 bacteria-positive spores could be collected in each round for both B+ and B++ populations, with exceptions where only 50,000 spores were available due to logistical constraints during FACS sessions (injection 3 Round 4 B++, and Round 15 B+ and B++). Each “round” constitutes one complete cycle of spore collection, growth to maturity, sporulation, and harvesting of the resulting spore population. The germination rates were determined by sorting single spores into 96-well plates and subsequent optical observation whether a mycelium formed using a stereomicroscope, and bacterial presence was*

routinely verified for a subset of FACS-sorted spores in each round. We have added this methodological explanation to the revised Methods section.

In section “Test against nutrient starvation and bacterial clearance”, what is meant by cryo-preserved? Spores preserved at -20C or -80C? Also, the title of this section refers to “bacterial clearance”, but it is not explained or detailed in the section.

Cryo-preservation refers to spores at -80°C in 50% (v/v) glycerol. We have clarified this storage condition in the revised Methods section.

Regarding bacterial clearance, this was assessed by examining the presence or absence of bacteria in germlings following nutrient starvation using fluorescence microscopy at 10X to 40x magnification on an inverted optical microscope. We have revised the section title and content to better explain that “bacterial clearance” refers to the loss of bacterial symbionts from fungal hosts under starvation conditions, and have provided methodological details for how this clearance was detected and quantified (lines 479 to 493)

Line 483: please clarify that this Rhizopus assembly is available in NCBI. Was it sequenced in the course of the current study? Is this the same fungal strain as used in the current study?

The Rhizopus assembly (GCA_964213375.1) is publically available in NCBI and was not sequenced as part of the current study. This assembly and gene annotations were generated in our previous work⁵ and represent the same fungal strain used in the current study. We have updated the Methods section to clarify the source and availability of the genomic resource.

Line 486 – where is the Rhizopus proteome coming from?

The Rhizopus proteome was obtained by extracting protein sequences from the genome assembly GCA_964213375.1 and submitting them to STRING-db using their 'Add organism' feature. The gene annotation of the assembly was described in our previous work⁵, where genes were predicted using BRAKER (v3.0.6) with the --fungus flag, and subsequently functionally annotated using eggNOG-mapper (v2.1.12) with the --target_taxa Fungi option. We have added this clarification to the Methods section to specify the source and processing of the proteome data.

Reviewer #1 (Remarks on code availability):

I have checked the provided code, which is openly available. Unfortunately, I do not have the relevant coding experience especially with image analysis to test its reproducibility.

Reviewer #2 (Remarks to the Author):

The research work submitted by Gassler and collaborators utilized fluid force microscopy to introduce a free-living bacterial strain, *Ralstonia pickettii* K-288, into symbiotic and asymbiotic strains of *R. microsporus*, which the authors refer to as EH and NH, respectively.

Authors hypothesized that implantation of this bacterium in the fungal cytosol will initially be perceived as a pathogen by the fungus, but when positive selection is maintained, this interaction will transition to a more stable association.

To test this hypothesis, the authors first labeled *R. pickettii* K-288 with mCherry to determine if this bacterium could colonize EH, cured EH, or NH strains of *R. microsporus*. They also tested if co-infection of *R. pickettii* and *M. rhizoxinica*, the natural bacterial symbiont of *R. microsporus*, was possible by co-cultivating cured EH and the two labeled bacterial strains (mCherry and GFP-labeled, respectively). These experiments showed that *R. pickettii* was unable to colonize EH or NH strains naturally. Colonization was also not possible when co-cultivation with *M. rhizoxinica* was performed in the cured EH fungus. Did the authors notice any differences in the “external” interaction of *R. microsporus* EH and NH towards *R. pickettii*? Can one speak of *R. pickettii* as a pathogen of *R. microsporus*? Why? What happens to the EH and NH fungus in the external presence of *R. pickettii*?

We thank the reviewer for the detailed evaluation and clarifying questions. To better assess external bacterial effects, we conducted a co-culture experiment on solid medium assessing the impact of external R. pickettii exposure on fungal colony growth of the EH and NH strain (Figure R2 a). External presence of R. pickettii reduced fungal growth in both the NH and EH strains. Although M. rhizoxinica also reduced fungal growth, the effect was more pronounced upon exposure to R. pickettii. In contrast, co-cultivation with E. coli did not impair fungal growth to a comparable extent. Notably, the NH strain showed greater sensitivity to bacterial challenge than the EH strain, an effect visible at the macroscopic level (Figure R2 b). These data suggests that R. pickettii exerts inhibitory or competitive effects on fungal growth even without establishing intracellular colonization. Beyond this characterization, a more detailed macro- and microscopic analyses during the infection assays was challenging due to the three-dimensional nature of fungal growth in liquid culture and bacterial proliferation in the surrounding media.

While these data provide evidence for compromised growth of the fungus in the presence of externally applied R. pickettii, we believe they extend beyond our manuscript's focus on intracellular colonization. We have instead prioritized additional qPCR and microscopy data that directly address the pathogenic character of R. pickettii in intracellular interactions (see below), which aligns with our primary objectives. However, we would be prepared to include the co-culture data set in the Supplementary Information if the reviewer considers it would enhance the manuscript.

Figure R2: Effect of extracellular exposure of *Ralstonia pickettii*, *Mycetohabitans rhizoxinica* and *Escherichia coli* on growth of the NH and EH strain. To assess the effect of extracellular exposure on fungal growth the different bacteria were cultured in nutrient broth at 28°C to stationary phase. One OD unit of cells was washed and resuspended in 1 mL HEPES2 buffer and 5 μ L of this suspension was spotted onto the center of nutrient broth agar plates supplemented with 1% (w/v) glycerol. Plates with *E. coli* and *R. pickettii* were pre-incubated at 16°C overnight; plates with *M. rhizoxinica* at 28°C for two days, to ensure bacterial growth at the time of fungal inoculation. Fungal spores from both the endosymbiont-harboring (EH) and the naive host (NH) strains were germinated overnight in 5 mL PD broth in WillCo dishes at room temperature. Individual germlings were transferred by pipette directly onto the bacterial spot under a 10x objective using a 10 μ L pipette, with successful placement verified immediately by stereo microscopy (Zeiss SteREO Discovery.V8). Plates were incubated at 28°C in the dark. Fungal colony diameters were measured daily until full plate coverage, with images taken at each time point and at the endpoint for all conditions. **a**, Growth curves show colony diameters for NH (dashed lines) and EH (solid lines) strains co-inoculated with *R. pickettii* (NH_Rp, EH_Rp), *E. coli* (NH_Ec, EH_Ec), *M. rhizoxinica* (NH_Mr, EH_Mr) and controls without bacteria (NH_none, EH_none); each condition was replicated biologically five times. **b**, Representative images show full plates (bar = 1 cm) and close-ups of each colony at the last measurable time point.

To directly address the pathogenic character of *R. pickettii* in intracellular interactions, we examined intracellular growth more closely. When *R. pickettii* gains access to the fungal cytoplasm - which may occur in nature through wounding or other breach mechanisms - it exhibits markedly different behavior compared to the natural endosymbiont. This is evident from microscopy of the passaged B+ line compared to the natural endosymbiont *M. rhizoxinica* in both the EH and NH background which is added as Supplementary Figure 7 to the revised manuscript) and shown below (Figure R3).

To quantify this difference, we developed a qPCR method to determine bacteria-to-fungal gene copy ratios in mature mycelium (Supplementary Figure 7). The natural endosymbiont *M. rhizoxinica* maintains controlled colonization levels of approximately 1 bacterial copy per 900-2400 fungal actin copies in EH strains (one ACT1 copy in genome) and 1 per 2700-16000 in evolved NH strain (two ACT1 copies in genome) from our previous work (ref⁵). In stark contrast, *R. pickettii* colonization reached 1 bacterial copy per 2-15 fungal copies, corresponding to two orders of magnitude higher intracellular proliferation than the natural endosymbiont.

This excessive intracellular proliferation characterizes *R. pickettii* as a pathogen-like organisms of *R. microsporius*. Specifically, the growth to high bacterial loads in our system is reminiscent of hyperproliferation of established intracellular pathogens in the host cytosol⁶⁻⁸, suggesting that *R. pickettii* exploits the fungal cellular environment for uncontrolled replication rather than

establishing the balanced relationship seen with *M. rhizoxinica*. We have added these additional findings that underline the pathogenic character of *R. pickettii* to the fungus to our revised manuscript (see changes in the result section, lines 165-172 and in the discussion, lines 290-299).

Figure R3: The colonization of the fungus by *Ralstonia pickettii* exceeds the one by *Mycetohabitans rhizoxinica*. a-b, pictures of young germlings of cultivated in liquid potato dextrose broth (PDB) medium showing that colonization of the EH strain (a) and the NH strain (b) by the natural endosymbiont *M. rhizoxinica* is apparently lower than by *R. pickettii* after germination (c); e-f, images of mature mycelium originating from single positive spores of EH colonized by its natural endosymbiont *M. rhizoxinica* (d), NH with *M. rhizoxinica* (e) and NH with *R. pickettii*. g, bacterial colonization obtained by qPCR shows that *R. pickettii* (recovered round 19 B_{pos} spores) reaches larger colonization levels than *M. rhizoxinica* in the EH and EH strain, shown are averages of log₁₀ transformed copy number ratios of fungal to bacterial copies from 9 (Evolved NH x Mr) to 10 (EH x Mr GFP and R19r B_{pos}) biological replicates ± standard deviation, low values indicate high bacterial colonization.

Using fluid force microscopy, authors injected labeled *R. pickettii* into NH *R. microsporus* (3 independent events) and selected spores that harbor either a low (B+) or a high bacterial load (B++) for five and up to 20 rounds of positive selection. The authors analyzed the fitness index, calculated as the product of the germination rate (i.e., germination success or percentage germination) and the fraction of spores that harbor *R. pickettii*. They noticed that this index changes with serial passaging. These experiments do not appear to reach stability, and the fitness index remains relatively low in both scenarios (high and low bacterial load). This data is based only on injection 3, which was the only one followed and analyzed until round 20. Authors should also analyze the outcomes of the two other events to round 20, thereby better supporting their claims. Do the three independent injections lead to similar results in terms of fitness?

We appreciate this comment regarding the analysis of all three injection lines to round 20. We acknowledge that examining all three lines to round 20 would have provided a more comprehensive dataset. However, we were constrained by limited access to the core facility during the experiment period, which prevented us from following all three lines using both B+ and B++ selection regimes in parallel. Extending the two additional injection lines to round 20 would require significant experimental commitment, amounting to approximately 20 weeks for labor time, including cultivation, weekly preparation, FACS sorting and imaging. We also would need to replicate line 3 as a control, as we would now need to start from frozen stocks. We would then

need to repeat the transcriptomics and data analysis for rounds 10 and 20. In total, this would take about 10-12 months, and is beyond the scope of this study.

However, all three independent injection lines showed consistent phenotypic trends through round 5, demonstrating reproducibility of the initial evolutionary dynamics across biological replicates. This consistency suggests that the fundamental processes we describe are not artifacts of a single injection event.

Additionally, when authors use positive spores to generate new ones, they redo the sorting. How many fresh spores are classified again as positive after this selection (percentage of the total)? Is this high? Was it different for R2, R11 and R20?

When regenerating spores from frozen stocks, we subjected spores from each respective round to FACS sorting using the same criteria and plating conditions as the original passaging experiment. The percentage of positive spores after re-sorting remained consistent with those observed in the original rounds (approximately 1% for B+ R2, 3% for B+ R11 and 4% for B+ R20). This consistency demonstrates that the bacteria positive phenotype was maintained during freezing and storage.

We typically sorted more than 10,000 spores, subsequently plating these under identical conditions to those used during the passaging experiment. However, we observed substantial variability in post-freezing viability across different rounds, with early adaptation rounds (R2) showing reduced germination rates after freezing compared to those from later rounds (R11, R20). Consequently, only a fraction of the positive spores successfully germinated after plating. Following germination, we harvested the spores and performed FACS to select bacteria-positive spores for subsequent imaging experiments. The proportions of positive spores obtained from these germinated populations aligned with the corresponding original rounds, confirming that the selection pressure had been maintained.

With spores from injection 3, the authors further analyzed that fungal growth decreased when *R. pickettii* colonized (NH vs R20) and that germination success tended to increase from R2 to R20, having spores from R20 a higher bacterial load. This result contrasts with previous research using *M. rhizoxinica* and NH *R. microsporus*, which found that an important adaptation to increase host fitness was to reduce or control bacterial load.

*We agree that the dynamics observed in the two studies differ. In our previous work with *Mycetohabitans rhizoxinica*⁵, the fungal host adapted quickly over 10 selection rounds by reducing the bacterial load per spore and enhancing its fitness. In contrast, the current study with *Ralstonia pickettii* shows that the adapted spores harbored an increased bacterial load, which correlated with a lower fungal growth rate. We hypothesize that this elevated bacterial burden results from an attenuated fungal defense response, which is also consistent with the lack of septa formation that we had observed with *E. coli* injections in our earlier work⁵. *R. microsporus* appear to tolerate this higher bacterial load, as evidenced by maintained germination success and spore viability despite the increased *R. pickettii* colonization. To clarify these contrasting observations, we have included a dedicated section in the discussion that contextualizes these findings with respect to our earlier research (see lines 347-349).*

A propo: When comparing B+ vs. B++, what is the average of bacterial cells contained per spore in each case?

Due to the clustering of the cells inside the spores, it is not possible to determine the exact number of bacteria per spore using optical and fluorescence microscopy. However, using an adapted script from one of our earlier studies (ref⁹), we quantified the bacterial volume in the B+ and B++ subpopulations as a proxy for bacterial abundance (see updated Figure 1 j). In addition, we have now included additional high-resolution images of B+ and B++ spores in Supplementary Figure 4, highlighting the colonization pattern upon vertical transmission. This volumetric analysis captures

the quantitative differences between the B+ and B++ populations that formed the basis for our selection regime.

Later, the authors studied the fungal transcriptomic response to the intracellular interaction with *R. pickettii*, showing that the transcriptional response between R20pos and NH was more similar than between R10pos and NH, suggesting that the fungus adapted its response to the symbiotic bacterium. Additionally, when authors compared the R20pos vs. R10pos, they identified up-regulated genes in the early round versus those in the late round, indicating the transitions likely required for the establishment of endosymbiosis.

One prominent aspect that authors did not consider in their work is the response and changes that occur to the bacterium during the serial passaging experiment. Therefore, the claim that the interaction changed from an antagonistic to a commensal interaction is not yet fully supported. For this, we need to know the changes suffered by the bacterium at both the genomic and/or transcriptomic levels.

We acknowledge that our assessment of the transition from antagonistic to commensal interactions is based primarily on phenotypic observations from the fungal perspective. We observe that the initial interaction is antagonistic, as evidenced by fitness deficits in the host fungus accompanied by a distinct transcriptomic signature indicative of activated defense mechanisms. As the fitness deficits decrease over successive rounds of passaging, the interaction appears to transition into a more commensal phase, where the bacteria continue to benefit while harm to the host decreases. We recognize that this interpretation reflects changes observed primarily from the host's perspective and that a complete understanding would require characterizing both interacting parties.

For the molecular characterization, we made a deliberate choice to focus on the fungal transcriptomic response. Revealing co-occurring transcriptomic changes in both the bacterial and fungal partners would require a dual RNA sequencing workflow tailored to our model system. During the experimental design phase, we carefully considered this approach and chose to focus on characterizing the host response as a foundational step before progressing toward more complex analyses of cross-kingdom communication in future work.

In response to the reviewer's comment, we have revised the manuscript to clarify that our conclusions about interaction dynamics are based on the fungal perspective and should be interpreted within this context. We have also carefully adjusted our terminology throughout the manuscript to reflect this limitation and improve clarity (see lines 81, 198 and 347-349).

In summary, the presented evidence suggests a host-driven physiological plasticity response rather than true evolutionary co-adaptation. Authors do not report heritable mutations in either fungus or bacteria nor are there functional adaptations suggestive of a transition toward stable endosymbioses, such as gene loss or intracellular specialization. In this regard, it is advisable to use terms such as “induced intracellular association” rather than “endosymbiosis” or “co-evolution” unless authors provide bacterial genomic or transcriptomic data to support a deeper adaptive process emerging through serial passaging.

We agree with the reviewer's distinction between physiological plasticity and evolutionary co-adaptation. Our data indeed demonstrates host-driven physiological changes rather than evidence for heritable mutations or functional adaptations typically associated with stable endosymbioses, such as gene loss or intracellular specialization. In response to this feedback, throughout the revised manuscript, we have carefully adjusted our terminology to reflect this distinction. We now use more precise terms such as "induced intracellular association," "physiological adaptation," and "host accommodation" rather than "endosymbiosis" or "co-evolution" to accurately describe the observed phenomena. We have also clarified that our findings represent host plasticity in response to bacterial colonization rather than co-evolutionary processes. See for example lines 281 and following for these terminological revisions.

Minor comments:

Abstract

Lines 6-7, "bacteria implantation" replace with "bacterial implantation"

We changed the respective sentence to "Following the implantation of bacteria into the cytosol,...", to stay consistent with our previous work⁵.

Line 7 compromised host fitness, as evidenced by reduced fungal viability (add a coma)

Done.

Line 24 "predispose them for" replace with "predispose them to"

Done.

Line 25 "Such adaptations, may provide" (remove the coma)

Done.

Line 31 "remains limited" replace with "remain limited"

Done.

Results

L85. Cured strain' is mentioned without a definition

We added a sentence explaining that "cured host" refers to the EH strain after removal of *M. rhizoxinica* by serial plating on PDA plates containing ciprofloxacin.

Line 125, "designed an serial" replace with "designed a serial"

Done.

Line 194, the hyphen in "NH-strain" should be consistent: either use "NH strain" without the hyphen or always use "NH-strain."

We will streamline and use "NH strain" throughout.

Discussion

Line 257 "symbiotic integration" replace with "symbiotic integration"

Done.

Lines 275-6 "pathogencontaining" replace with "pathogen-containing"

Done.

Methods

Line 419, 100x magnification (000x 1.45 CFI Plan Apo Oil objective). Its 000x or 100x?

Thank you for the comment and spotting this typo. It refers to 100x magnification.

Sup_Video_Legends: Supplementary Video 2 label is repeated. The last one should be Video 3. Beautiful videos!

Thank you for the comment. The video legends will be corrected and changed.

Reviewer #2 (Remarks on code availability):

I did not have the time to test the code.

I only checked it was available as stated.

Reviewer #3 (Remarks to the Author):

This is a novel approach to reveal how initial negative influence of endofungal bacteria evolves towards more commensalism. Although the described experimental system might have limited time frame, this provides a simplistic and excellent approach to study such evolutionary transition that might exist in nature. The described data are detailed and thorough, I have no major concerns. However, the writing of the manuscript could benefit from slight attention to describing the various strains and treatments. I understand the author aimed for a condensed text, but when I took down the manuscript for few hours, I was unable to follow all abbreviations. Therefore, I suggest revising the text to describe not only the treatments with abbreviations but also add half a sentence on what was compared (e.g. in line 206-207, I am lost what is compared with what).

We thank the reviewer for the encouraging words on the manuscript and constructive feedback. We have carefully revised the entire manuscript to improve readability and clarity, with particular attention to sections involving abbreviations and the description of treatments and strains. Specifically, we have: (i) added clearer explanations of strain designations upon first mention (e.g. line 124 for definition of B_{pos} , B+ and B++), (ii) included brief descriptive phrases alongside abbreviations to remind readers of what is being compared (e.g. lines 251-252 explaining temporal comparison of B_{pos} transcriptomic samples), (iii) expanded treatment descriptions to specify what conditions are being contrasted (e.g. lines 224-226 for explanation of R10 B_{pos} and R20 B_{pos}).

Also, I am not fully convinced that fungus displays a triggered immune response, but it might be a stress induced reaction by the fungus. I understand immune response as specific reaction to an immunogen performed by specialized cells or protein pathway, but I only see here stress response being induced in the fungus, no specific protein pathway is induced. Thus, I suggest the authors to remove the term immune response, unless they add additional data describing an immune response like pathway (e.g. line 8, 80, 290, 292). The authors nicely avoid this term in the rest of the manuscript, which I agree with.

We thank the reviewer for this comment regarding the use of the term "immune response." We agree that the term should be used with care, particularly in fungi, where the definition of an immune system is less well-established than in animals or plants¹⁰⁻¹².

*Our intention was not to imply the presence of a highly specific, receptor-mediated immune pathway (which we indeed do not have evidence for), but rather to highlight that the fungus exhibits a transcriptional response to intracellular bacteria that resembles a general defense response in *R. microsporus*^{12,13}. We agree with the reviewer that this is more accurately described as a general stress response.*

We hypothesize that this stress response could represent a component of a broader fungal defense system, conceptually similar to stress responses that contribute to plant immunity^{14,15}. To reflect this and to address the reviewer's concern, we have revised the manuscript to avoid the term "immune response" in the respective lines, instead using terms such as "stress response" or "defense response".

Finally, the presence of microcompartment coupled with host-mediated functional segregation is very intriguing idea, and such should be easily visible with using high resolution microscopy or using fluorescent membrane stains. Especially comparing the initially introduced bacterial cell localization and those after adaptation would make the story more compelling if localization is indeed altered.

*We agree that compartmentalization of *R. pickettii* over the course of adaptation is an intriguing observation.*

Immediately after injection, bacteria are observed to replicate freely in the cytosol, without evidence of vesicular encapsulation (we have included an additional supplementary video showing cytosolic growth after injection, Supplementary Video 1 in revised manuscript). In contrast, in germlings from passaged spores, we observe altered localization suggestive of host-derived compartmentalization (as shown in adapted Supplementary Video 3).

*While these observations suggest potential changes in bacterial localization during adaptation, we do not currently have the temporal imaging data to determine whether these observations are driven by adaptation and/or bacterial load, which represents an important direction for future research. Moreover, standard membrane stains are not reliable in *R. microsporus* due to cell wall impermeability and protocols for such labeling not yet being established. While electron microscopy could provide additional clues, it falls outside the scope of the current work but represents a promising avenue for future studies.*

Line 265: I am not fully convinced about “co-evolved compatibility traits”. It should be explained better what is meant under this term and whether this is specifically demonstrated. No specific molecular mechanisms is revealed, so I am not sure what this could refer to.

*We removed the term “co-evolved compatibility traits” and changed the sentence to “The ability of *R. pickettii* to proliferate robustly within the fungal mycelium following injection, despite lacking a history of co-evolution with the host, highlights its potential for opportunistic intracellular persistence.” to more clearly describe the observed phenomenon.*

Line 307: Is it really pathogen if host is not killed, but delayed for growth? Also, the bacterial entrance would not happen without active introduction of those cells. Thus, this is not pathogenesis but possibly a defector. I understand the need of commonly used terms, but the traditional pathogen term might be misinterpreted here.

*We agree that the interaction between *R. pickettii* and *R. microsporus* does not fit to classical host-pathogen relationships. However, pathogenicity does not necessarily require host killing, but rather a measurable fitness cost to the host that benefits the invader, often through evasion of host defense^{16,17}. A close relative of the *R. pickettii* K-288 type strain was reported as a stable endosymbiont of *Rhizopus*; however, the natural route of infection remains unknown¹⁸. One of the strengths of our experimental system is the ability to bypass natural entry barriers and directly test intracellular persistence and proliferation. In this controlled context, *R. pickettii* imposes a fitness cost on the host and displays characteristics of opportunistic pathogen-like behavior. To substantiate the antagonistic behavior, we have quantified bacterial growth within the fungal mycelium and added these data to the manuscript (Supplementary Figure 7).*

References

1. Luna, E., Bruce, T. J. A., Roberts, M. R., Flors, V. & Ton, J. Next-Generation Systemic Acquired Resistance. *Plant Physiol.* **158**, 844–853 (2012).
2. López Sánchez, A., Pascual-Pardo, D., Furci, L., Roberts, M. R. & Ton, J. Costs and Benefits of Transgenerational Induced Resistance in Arabidopsis. *Front. Plant Sci.* **12**, 644999 (2021).
3. Yabuuchi, E., Yano, I., Hotta, H., Nishiuchi, Y. & Kosako, Y. Transfer of Two Burkholderia and an Alcaligenes Species to Ralstonia Gen. Nov.: Proposal of Ralstonia pickettii (Ralston, Palleroni and Doudoroff 1973) Comb. Nov., Ralstonia solanacearum (Smith 1896) Comb. Nov. and Ralstonia eutropha (Davis 1969) Comb. Nov. *Microbiol. Immunol.* **39**, 897–904 (1995).
4. Moebius, N., Üzümlü, Z., Dijksterhuis, J., Lackner, G. & Hertweck, C. Active invasion of bacteria into living fungal cells. *Elife* **3**, e03007 (2014).
5. Giger, G. H. *et al.* Inducing novel endosymbioses by implanting bacteria in fungi. *Nature* **635**, 415–422 (2024).
6. Knodler, L. A. *et al.* Dissemination of invasive Salmonella via bacterial-induced extrusion of mucosal epithelia. *Proc. Natl. Acad. Sci. U. S. A.* **107**, 17733–17738 (2010).
7. Chong, A., Starr, T., Finn, C. E. & Steele-Mortimer, O. A role for the Salmonella Type III Secretion System 1 in bacterial adaptation to the cytosol of epithelial cells. *Mol. Microbiol.* **112**, 1270–1283 (2019).
8. Birmingham, C. L. *et al.* Listeriolysin O allows Listeria monocytogenes replication in macrophage vacuoles. *Nature* **451**, 350–354 (2008).
9. Gäbelein, C. G., Reiter, M. A., Ernst, C., Giger, G. H. & Vorholt, J. A. Engineering endosymbiotic growth of *E. coli* in mammalian cells. *ACS Synth. Biol.* **11**, 3388–3396 (2022).
10. Daskalov, A. Emergence of the fungal immune system. *iScience* **26**, (2023).
11. Gaspar, M. L. & Pawlowska, T. E. Innate immunity in fungi: Is regulated cell death involved? *PLOS Pathog.* **18**, e1010460 (2022).
12. Pawlowska, T. E. Symbioses between fungi and bacteria: from mechanisms to impacts on biodiversity. *Curr. Opin. Microbiol.* **80**, 102496 (2024).
13. Lastovetsky, O. A. *et al.* Molecular dialogues between early divergent fungi and bacteria in an antagonism versus a mutualism. *MBio* **11**, 1–19 (2020).
14. Maier, B. A. *et al.* A general non-self response as part of plant immunity. *Nat. Plants* **2021** **7**, 696–705 (2021).
15. Keppler, A. *et al.* Plant microbiota feedbacks through dose-responsive expression of general non-self response genes. *Nat. Plants* **11**, 74–89 (2025).
16. Montarry, J., Hamelin, F. M., Glais, I., Corbi, R. & Andrivon, D. Fitness costs associated with unnecessary virulence factors and life history traits: Evolutionary insights from the potato late blight pathogen *Phytophthora infestans*. *BMC Evol. Biol.* **10**, 1–9 (2010).
17. Diard, M. *et al.* Stabilization of cooperative virulence by the expression of an avirulent phenotype. *Nature* **494**, 353–356 (2013).
18. Itabangi, H. *et al.* A bacterial endosymbiont of the fungus *Rhizopus microsporus* drives phagocyte evasion and opportunistic virulence. *Curr. Biol.* **32**, 1115–1130.e6 (2022).

REVIEWERS' COMMENTS

Reviewer #2 (Remarks to the Author):

The authors did a great job in answering the concerns we reviewers had and also in successfully incorporating most of the major recommendations.

Thank you for the encouraging assessment and thorough comments.

I thank authors for investigating in more detail the effects of the external interaction between *R. microsporus* (EH and NH) and *R. pickettii*. and also for comparing the interactions with *M. rhizoxinica* and *E. coli*. I agree that the focus of the manuscript is the intracellular interaction of *R. microsporus* NH, a naturally asymbiotic fungus, with *R. pickettii*, which is not a natural symbiont of *R. microsporus*. Thus, the generated information is valuable, but not needed to understand the main story of this work.

Thank you.

I also highly value the efforts to estimate the abundance of the bacterial symbionts in the fungal host using qPCR. From this data, it is pretty clear that *R. pickettii* cell numbers within fungal NH mycelia are much higher than those achieved by *M. rhizoxinica*. A propo supplementary Figure 7 it says "(g), bacterial colonization obtained by qPCR shows that *R. pickettii* (recovered round 19 Bpos spores) reaches larger colonization levels than *M. rhizoxinica* in the EH and EH strain", and it should be "in the EH and NH strain". A small typo.

Corrected.

Some other changes I still suggest are:

Change the label "Germination rate" for "Germination success" or "Germination" as in your previous report (Giger et al., 2024). This change is required in figures and text along the whole manuscript (for example in Figures 1j, 2b, 2e, 3c, S5, S6, etc). This change will increase the consistency and accuracy of the work, as authors report the percent of spores that did germinate from the total in each round, and not the rate of germination (spores germinated by a unit of time (min, hr, day)).

Changed throughout to "Germination success".

In Fig. 2f the label for the y-axis should be "Positive fraction (%)" instead of "Positive rate(%)", right?

Corrected to "Positive fraction (%)".

I wonder why authors use different scales for Figure 2b (0, 20, 40, 60,..) and 2e (0, 1, 10, 100) if they show germination (%) in both cases. I think it might be easier for the reader to compare numbers if the same scales are use in both panels.

Figure 2e uses a logarithmic scale to better visualize low-end values and relevant comparisons. We now label the axis "log germination success (%)" for clarity.

Finally, I am deeply curious about the potential changes that might have occurred to the bacterial genome after 20 rounds of positive selection, and under which circumstances (ecological context) this novel symbiosis would further stabilize. Great work that opens many questions and new possibilities of testing them!

Thank you. We agree these are fascinating directions for future research directions.

Reviewer #2 (Remarks on code availability):

Only that it is publicly available, but I did not have the time to test if it works.

Reviewer #3 (Remarks to the Author):

Thank you for adjusting the manuscript according to the suggestions, I have no further comments. Congratulations on the superb work!

Thank you for the positive assessment and efforts during the review process.